# Nanomechanical DNA resonators for sensing and structural analysis of DNA-ligand complexes

Stefano Stassi[1,5], Monica Marini[1,2,5], Marco Allione[2], Sergei Lopatin[3], Domenico Marson [4], Erik Laurini [4], Sabrina Pricl[4], Candido Fabrizio Pirri[1], Carlo Ricciardi [1] & Enzo Di Fabrizio[2]

The effect of direct or indirect binding of intercalant molecules on DNA structure is of fundamental importance in understanding the biological functioning of DNA. Here we report on self-suspended DNA nanobundles as ultrasensitive nanomechanical resonators for structural studies of DNA-ligand complexes. Such vibrating nanostructures represent the smallest mechanical resonator entirely composed of DNA. A correlative analysis between the mechanical and structural properties is exploited to study the intrinsic changes of double strand DNA, when interacting with different intercalant molecules (YOYO-1 and GelRed) and a chemotherapeutic drug (Cisplatin), at different concentrations. Possible implications of our findings are related to the study of interaction mechanism of a wide category of molecules with DNA, and to further applications in medicine, such as optimal titration of chemotherapeutic drugs and environmental studies for the detection of heavy metals in human serum.

[1] Dipartimento di Scienza Applicata e Tecnologia, Politecnico di Torino, Corso Duca Degli Abruzzi, 24, 10129 Torino, Italy. [2] Physical Science and Engineering and BESE Divisions, King Abdullah University of Science and Technology, Thuwal 23955-6900, Saudi Arabia. [3] Imaging and Characterization Core Lab, King Abdullah University of Science and Technology, Thuwal 23955-6900, Saudi Arabia. [4] Molecular Biology and Nanotechnology Laboratory (MolBNL@UniTS) – DEA, University of Trieste, Piazzale Europa 1, 34127 Trieste, Italy. [5] These authors contributed equally: Stefano Stassi, Monica Marini. Correspondence and requests for materials should be addressed to C.R. (email: carlo.ricciardi@polito.it) or to E.D.F. (email: Enzo.DiFabrizio@KAUST.EDU.SA)

Cellular processes at the molecular level can be severely affected by variations in the pristine conformation of the nucleic acids. For example, intercalants and chemotherapeutic analytes are responsible for different molecular variation into nucleic acid molecules.

As per definition, an intercalant is a planar aromatic compound able to insert itself between two adjacent DNA base pairs[1]. The intercalation binding obeys the rule of nearest-neighbor exclusion[2]. There is a wide choice of intercalants, widely used as stains to visualize DNA in molecular biology protocols[3]. DNA fluorescent stains show a very weak emission when free in solution, whereas their fluorescence increases to several orders of magnitude when intercalated in DNA. Intercalating agent action depends on molecules chemistry and on the binding site at the double helix.

The interaction between double-stranded nucleic acids and bis-intercalating molecules such as YOYO-1 or GelRed and the chemotherapeutic molecule cisplatin (CisPt) are at the origin of several DNA/ligand complexes through the intercalation mechanism. Intercalation occurs between adjacent base pairs, like in the case of ethidium bromide[4], YOYO-1 and GelRed bis-intercalate, while DAPI and other stains (mainly used in cellular biology) interact with the nucleic acids minor groove only. In the mentioned cases, the chemical interaction induces severe DNA variations: size, charge, stability, solution viscosity, angle of inclination between the phosphate groups, unwinding in the site of intercalation, and elongation of the helix due to the increase in contour length[5,6]. This alteration in the pristine conformation of the nucleic acids significantly perturbs their biological function and consequently the interaction with molecular machines by interfering with the enzymatic-mediated fundamental cellular processes, such as reparation, replication, or transcription[7]. Bis-intercalation pattern is expected to occur each four bases per intercalator molecule. The lengthening of the molecule causes changes in the density of the nucleic acid, decreasing from 1.7 g $cm^{-3}$ to ~1.5 g $cm^{-3}$[8,9].

Cisplatin (CisPt) is an inorganic platinum-derived agent (cis-diaminedichloroplatinum) that interacts with nucleophile molecules such as nucleic acids[10], producing platinum–DNA (Pt–DNA) adducts that have a central role in cancer treatment[11]. The interaction between the platinum agent and the DNA molecule gives rise to preferentially a covalent bond between the metal atom and the N7 of purine bases (adenine and guanine). This strong chemical variation can take the form of monoadducts, inter-, or intra-strand cross-links. The intra-strand cross-links occur mainly between adjacent bases on the same hemi-helix and constitute ~87% of the adduct (1,2-intrastrand d(GpG), 62%; 1,2-intrastrand d(ApG), 25%). 1,3-intrastrand cross-links d(GNG) between nonadjacent bases is only 10% and the percentage drops down for inter-strand cross-links and monofunctional adducts (down to 2% total). These alterations result in apoptosis and cell growth inhibition.

Resonance analysis of suspended mechanical structures has been widely implemented in the last few decades for extremely sensitive mass and force detection. Zeptogram mass resolution was obtained by shrinking the size of silicon resonators to the nanometric scale[12], while the implementation of 1D and 2D materials, such as carbon nanotubes or graphene membranes, increases the sensitivity down to the yoctogram range[13,14]. On the other hand, modal vibrational analysis was also implemented as a noninvasive indirect approach to investigate the mechanical properties of the inorganic nanostructure. This approach was proposed by Treacy et al. to measure Young's modulus of individual carbon nanotubes from thermally excited vibration[15]. This method was successfully applied to other 1D and 2D nanostructures, providing a fast and contactless technique to unveil their mechanical properties[16,17]. The mechanical properties of the DNA double helix differ from other biological or synthetic polymers, in terms of enhanced stiffness, higher torsional rigidity, and overwinding behavior under tension[18,19]. Recent studies of direct manipulation of DNA molecules with optical[20], magnetic[21], or mechanical tweezers[22], and atomic force microscope[23] underlined that these properties are strongly related to the helicoidal structure and the periodicity of the base sequence. It was demonstrated that the degradation of the periodic DNA structure, such as under therapeutic X-ray radiation, causes a noteworthy worsening of its mechanical properties[22]. Similarly, the intercalation of molecules into the double-helix structure or its local deformation is thus expected to directly impact the mechanics of the DNA stiffening or softening its structure, and thus changing its resonance spectrum.

In this work, we propose a method to investigate the binding of aromatic and platinized molecules to the DNA strand by evaluating the variation of mechanical properties of DNA–ligand complexes with a fast and contactless approach based on the measurement of the resonance frequency of suspended DNA bundles. The central idea of this work is that the structural effects induced by the intercalation mechanism can be measured through mechanical resonance oscillation of a DNA bundle suspended on a microfabricated super-hydrophobic substrate. The effective Young's modulus of the suspended DNA resonators is determined and related to the structural changes induced in the DNA-ligand complex. Furthermore, the effect of intercalant concentration is evaluated through titration experiments, suggesting possible applications in structural and biomedical studies, where molecular interactions are quantitatively determined by mechanical resonance and direct imaging measurements.

## Results

**Suspended DNA bundle characterization**. The suspended DNA bundles were prepared with a well-established approach reported in previous papers[24–26] (see the Methods section). The DNA bundle and the intercalation results were preliminarily checked by scanning electron microscopy (SEM) and by fluorescence microscopy. As reported in Fig. 1a, b, the bundles were suspended between pillars, over a wide area of the super-hydrophobic device used in this work. The fluorescence images gave an immediate response about the successful intercalation with GelRed and YOYO-1 as in Fig. 1c, d, confirming the interaction between DNA and dyes. Additional images of the intercalated DNA bundles, together with the sample treated with CisPt are reported in Supplementary Figs. 1–4.

The structural alterations of the DNA and in particular, the interbase distance changes, were studied by HRTEM. In Fig. 2, the direct images of DNA fibers in their pristine (A-DNA) and in the intercalated conformations are reported[24,27]. HRTEM micrograph analysis (Fig. 2) indicates a clear variation of the interbase distance of the A-DNA form due to the interaction with ligands, such as bis-intercalators and CisPt. The normal interbase distance of A-DNA at a relative humidity (RH) below 75% is of 2.7 Å[28,29] (Fig. 2a) and increases to ~4.5 Å with YOYO-1 and GelRed (Fig. 2b). The bis-intercalators saturate the DNA helices, placing themselves in each of four base pairs and causing the regular increase of the period with respect to DNA only. Panel c reports the effect of the CisPt/DNA adduct formation. The intra-strand cross-links cause severe alteration of the DNA helix, including bending and hydrogen bond disruption. The interbase distance becomes 5.2–5.7 Å, depending on the anchorage location of CisPt.

**DNA molecular dynamics simulations**. With the aim of understanding if the intercalation process could alter the

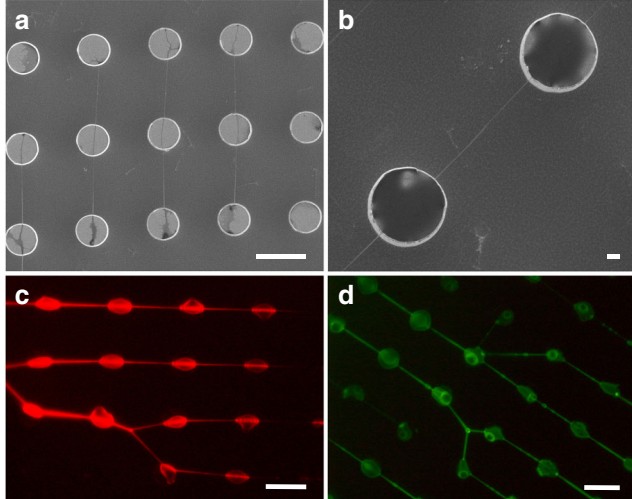

**Fig. 1** SEM and fluorescence microscope images of intercalated DNA. After the complete evaporation of the solvent (**a**), DNA bundles can be imaged by SEM: they are homogeneously suspended and distributed between silicon micro-pillars. In (**b**), a particular of a suspended DNA bundle, covering the pillar–pillar distance of 12 μm. DNA/intercalator optical images acquired using an excitation of (**c**) 540/25 nm (DM 565, BA 605/55) for the GelRed evaluation and of (**d**) 465–495 nm as in the case of YOYO-1. The scale bars correspond to 10 μm, except for (**b**) which corresponds to 1 μm

mechanical properties of the double-strand DNA, whose variations are detectable at the molecular level, we performed steered molecular dynamics (SMD) simulations under constant-force (CF) conditions[30] on a 66-base double-strand DNA in the absence and in the presence of the intercalant compounds YOYO-1 and CisPt, both with a well-established chemical formula. Notice that we did not include GelRed SMD because, at the moment of this drawing up, the chemical formula is unknown and never released by the company. In order to use a concentration condition similar to the experiments, we saturated the DNA with 16 molecules of YOYO-1 or seven molecules of CisPt. In the CF method, a constant-force vector is applied at one end of the DNA strand, while the other end was restrained. As it results in this uniaxial pulling force, the DNA elongates to a stable length, depending on the applied forces (Fig. 3a, b). At the same applied stresses, the three systems showed a different mechanical response, and the complete stress–strain plot allowed us to calculate the corresponding Young's modulus ($E_{SMD}$) for each complex. The three obtained stress–strain responses, in the range of the applied pulling force, have a linear behavior (Fig. 3b), and the slope of the linear fitting indicated the $E_{SMD}$ values, which are estimated to be 1.50 ± 0.11 GPa, 2.61 ± 0.18 GPa, and 1.11 ± 0.09 GPa for bare DNA and DNA intercalated with YOYO-1 and CisPt, respectively. Comparing the Young's modulus of pristine DNA with that of intercalated complexes, the DNA intercalated with YOYO-1 and CisPt showed a variation of its elastic properties. The intercalating molecules act as a structural reinforcement, stiffening the DNA structure. On the other hand, the presence of CisPt, covalently bounded to the DNA bases, led to a significant softening of the DNA structure, since the CisPt causes the local double-helix deformation. A plausible molecular rationale can be found by analyzing an important property that is a driving force of the molecular stability of the DNA helix: the hydrogen bond network among the bases. It is extremely worthy to note from Fig. 3d that the total number of the H bonds detected for pristine DNA and for DNA with YOYO-1 initially is

comparable, while at the end of the SMD simulation, the pristine DNA lost about 30% of the hydrogen bonds. On the contrary, the DNA intercalated with YOYO-1 kept this number almost stable after stretching. The intercalated YOYO-1 molecules show robust interactions with the DNA bases through π-stacking and electrostatic interactions (Fig. 2b), and this reinforces the stability of the DNA intramolecular H bonds. The result of this behavior is translated in the increase of the YOYO-1 sample structural stiffness. Regarding the effect of CisPt on the DNA H-bond network, it is clearly expected that at the beginning of the SMD simulation, we would have detected a lower H-bond number, compared with the bare DNA (Fig. 3d), since the CisPt adducts interrupt this network via a covalent bond with guanine N7 (Fig. 2c). Interestingly, our analysis showed a decrease in H-bond number of the DNA/CisPt complex, after stretching, of more than 40%. Furthermore, we point out that globally, even other DNA intramolecular H bonds become less stable with a consequent variation of the mechanical properties.

The mechanical properties of the DNA were then experimentally investigated by performing a vibrational analysis of DNA bundles. Indeed, the suspension of a single DNA filament, even if possible, suffers from mechanical stability, both during electron beam investigation in the TEM and laser examination in the vibrational analysis.

**Nanomechanical DNA resonator analysis.** The mechanical properties of suspended DNA bundles were investigated by analyzing their amplitude vibration response in the frequency domain using a Laser Doppler Vibrometer (scheme reported in Figs. 4 and 5a). The laser was focused roughly at the center of the bundles by the same objective used for collecting the reflected light, whose interference with the reference beam was used to evaluate the amplitude of the vibration (Fig. 5b, c) with a sub-picometer resolution[31]. This analysis is normally performed on micro- and nano-gravimetric sensors to investigate the resonance frequency shift induced by adsorption of chemical or biological analytes on the sensor surface[32,33]. Otherwise, it can be used to indirectly evaluate the mechanical properties of suspended structures from their resonance frequencies[16,17]. These DNA bundle oscillators, entirely composed of DNA, represent the smallest resonators fabricated with biological material. The length and the diameter of the studied resonators were in the 10–15 μm and 30–100 nm range, respectively. The diameter was quite homogeneous over the length of the bundles with a variability around 7% (details in Supplementary Note 2).

From the frequency spectra of the DNA resonators measured in air, it was possible to evaluate the frequency of the first three flexural resonance modes (Fig. 5c). As expected, the large hydrodynamic force in air acting on the bundle leads to a low-quality factor, but the resonance peaks are clearly measurable. The quality factor, which gives a quantification of the sharpness of the resonance peaks and thus of the damping of the resonators, computed as the resonance frequency divided by the bandwidth at 3 dB, was around 5 for the first mode and 8 for the second and the third. The quality factor of the DNA resonators in air is comparable with silicon nanowire oscillators[34], because damping is dominated by the interaction with the surrounding gas molecules. The same analysis was also performed in vacuum. The quality factor $Q$ of the DNA resonators was much higher (around 250), because gas damping can be neglected in the ballistic regime and the dissipation mechanisms are only related to clamping and intrinsic losses (Fig. 5d). Under such conditions, the DNA biological resonators exhibit a quality factor around one order of magnitude lower than that of inorganic ones ($Q$~2000 for silicon nanowires)[34]. On the other hand, the amplitude

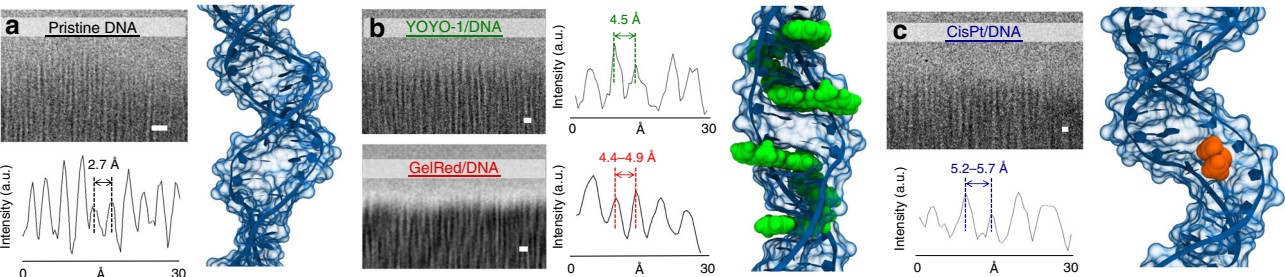

**Fig. 2** HRTEM images, related metrology, and schemes. Interbase distance details of a pristine DNA bundle (**a**), DNA bundle with bis-intercalators (**b**), and DNA with CisPt adducts (**c**). The DNA interbase distance changes from 2.7 Å in the pristine DNA conformation to ~4.5 Å and 5.2 Å in the DNA altered by the presence of intercalant molecules or CisPt, respectively. The scale bars correspond to 10 Å. Notice that for panels **b** and **c**, the models are just an indication of the action of each intercalant molecule and the changes of the periods are not reported

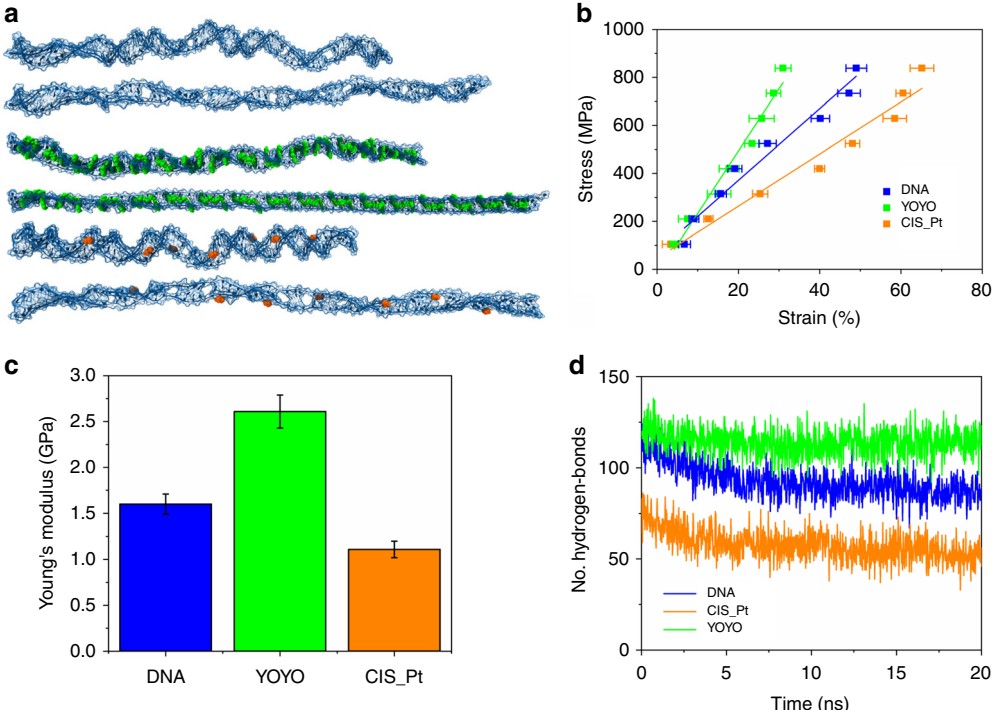

**Fig. 3** SMD simulations of pristine DNA and DNA intercalated with YOYO-1 and CisPt. **a** Simulated conformational structures of the bare DNA (top), DNA intercalated with YOYO-1 (center), and CisPt (bottom) under uniaxial stretching deformation. In each figure, the first corresponds to the initial structure, while the second represents the final conformation reached at the maximum simulated strain. **b** Stress–strain curve of the unidirectional traction applied to the DNA (blue), DNA/YOYO-1 (green), and DNA/CisPt (orange) systems. The Young's moduli ($E_{SMD}$) for each complex are calculated from the slope of the linear fitting. The strain at each force has been averaged over three simulations and the corresponding standard errors are reported. **c** Calculated Young's modulus ($E_{SMD}$) values for DNA (blue), DNA/YOYO-1 (green), and DNA–CisPt (orange) complexes. **d** Change in the number of hydrogen bonds of DNA (blue), DNA/YOYO-1 (green), and DNA/CisPt (orange) systems during the simulation time applying the maximum stress value. Source data are provided as a Source Data file

measurement in the vacuum chamber resulted considerably more difficult because of the background mechanical vibrations induced by the pumping system and by the intensity reduction of the reflected light from the resonators due to the partial absorption of the chamber glass cover. More interestingly, the vacuum analysis showed a splitting of the resonance mode peak, which was not resolved in air because of the low-quality factor. This phenomenon is related to the splitting of the flexural vibration modes of the resonator in two flexural modes vibrating on the orthogonal planes defined by asymmetries in the section geometry of the nanostructure, as observed in inorganic nanoresonators, such as silicon or GaAs/AlGaAs nanowires[34,35].

In a cylindrical resonator, for symmetry reasons, the two modes are identical and have the same frequency value, independently of the referred plane. However, if the resonator cross section is not perfectly circular, the symmetry is broken and the resonator vibrates in two orthogonal directions related to the maximum ($d_{max}$) and minimum ($d_{min}$) axes of the ellipse (corresponding to the maximum and minimum cross-section area moments, respectively), see Fig. 5e. Consequently, the splitting of the resonance peak in Fig. 5d evidences a deformation of the circular cross section toward an elliptical shape in our DNA resonators. The asymmetry factor $\Omega = (d_{max} - d_{min})/d_{min}$ can be calculated from the resonance frequencies of the split peaks as $\Omega = (f_f - f_s)/f_f$

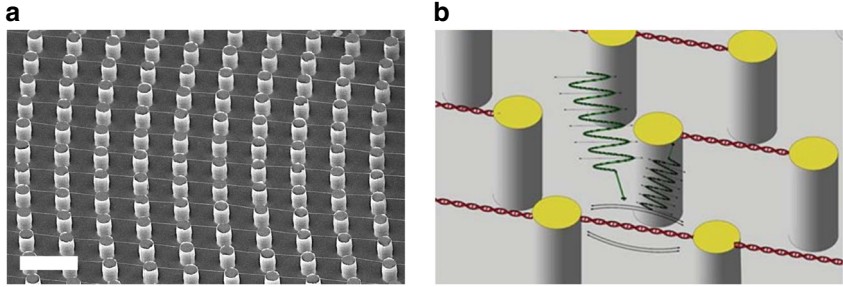

**Fig. 4** Representative DNA bundle and the mechanical vibrating scheme. **a** SEM image showing suspended bundles between pillars. **b** The cartoon symbolizes the mechanical vibration of the DNA bundle with the incoming and reflected laser beams used to detect the frequency and amplitude of DNA oscillation

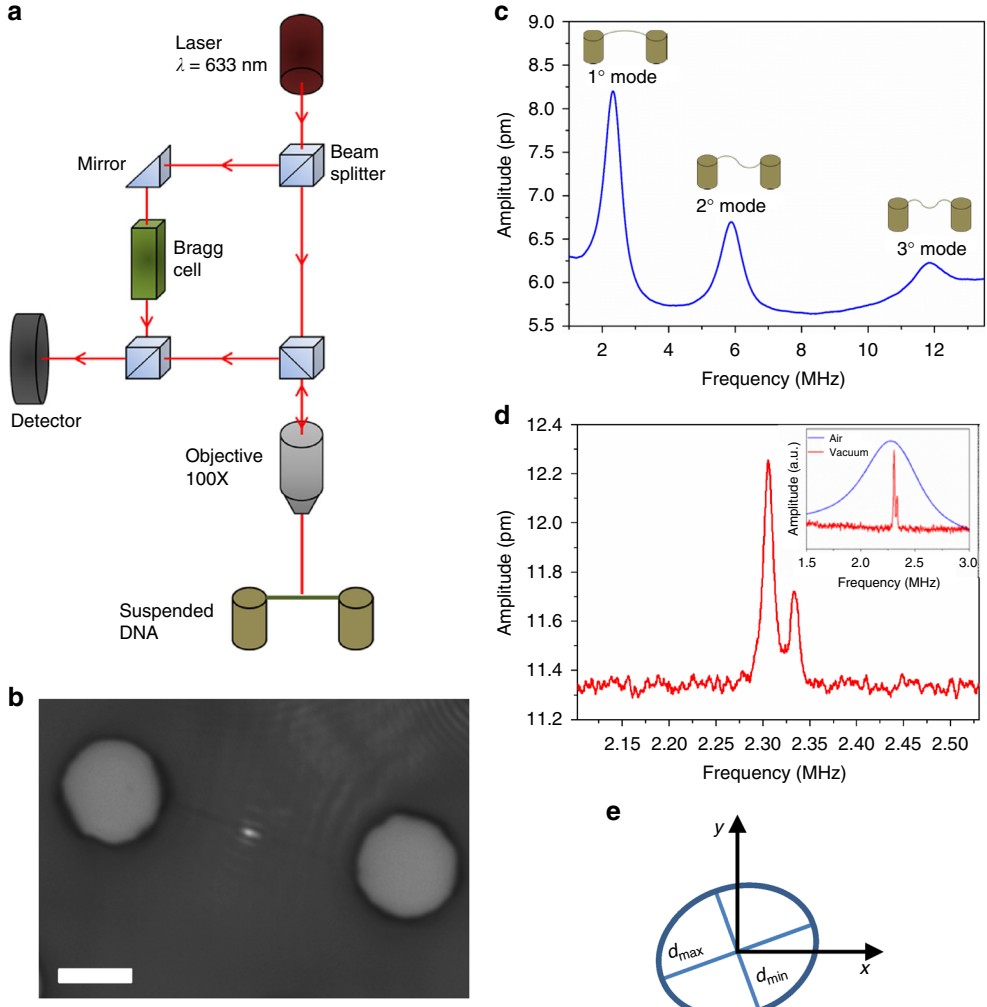

**Fig. 5** Nanomechanical characterization of suspended DNA bundles. **a** Scheme of the experimental setup composed of the Laser Doppler Vibrometer system. **b** Optical image of the device under test, showing the laser focused on a DNA bundle suspended between two adjacent pillars. The scale bar corresponds to 5 μm. **c** Vibration spectra of a DNA resonator measured in air, showing the first three resonance modes. A sketch of the mode shape is reported for each resonance. **d** Vibration spectra of a DNA resonator measured in vacuum, showing the splitting of the fundamental mode into two peaks. The splitting is due to the motion on the two orthogonal directions related to the axis of the maximum ($d_{max}$) and minimum ($d_{min}$) of the cross-section diameter. In the inset, the difference between the fundamental mode measured in air and vacuum can be clearly seen. **e** Schematic of the elliptical cross section of the DNA resonator. The variation between maximum ($d_{max}$) and minimum ($d_{min}$) of the cross-section diameter is exaggerated for a better understanding

where $f_f$ and $f_s$ represent the fast and slow resonance frequencies, respectively. From the analysis of the peak splitting in several DNA resonators, it results in an average value for $\Omega$ of 1.15%, corresponding to a variation between $d_{max}$ and $d_{min}$ of the bundle from 0.35 to 1.15 nm, i.e., less than half of the diameter of a single A-DNA strand[24].

The theoretical sensitivity of a resonator, defined as the change in resonance frequency due to a change in mass, represents a typical figure of merit of nanomechanical devices. The sensitivity can be computed as:

$$S = \frac{\delta f}{\delta m} = -\frac{f}{2m} \tag{1}$$

where $f$ and $m$ represent the resonance frequency and the mass of the DNA oscillators. From the analysis of the vibrational response of the DNA resonators the theoretical sensitivity resulted ranging from 4.2 to 9 Hz/ag depending on the length and diameter of the bundle. Considering the frequency stability of the resonator, the limit of detection of the DNA resonators is between 22 and 48 ag (details in Supplementary Note 3).

**Effect of the intercalant on the DNA mechanical properties**. A clamped–clamped resonator like our DNA bundle can be viewed in two different configurations, namely a beam configuration when the flexural rigidity effect (directly related to the material Young's modulus) is dominating, or in a string configuration when the bending vibration is governed by tensile stress effect[36–38]. The knowledge of the elastic configuration has a fundamental impact on the analysis of the intrinsic mechanical properties of the resonator and on all the considerations that can be done on the constitutive material. During the deposition process, when water evaporation occurs, the DNA bundles are pulled from one pillar to the proximal one. This mechanism could induce the presence of tensile stress in the resonators. Therefore, an analysis of the ratio of the resonance frequencies of the modes was first done, to have an indication for the presence of stress in the bundles.

Resonance frequencies $f_n$ of a clamped–clamped beam under tensile stress are defined as[38]:

$$f_n = \frac{\lambda_n^2}{2\pi L^2} \sqrt{\frac{EI}{\rho A}} \sqrt{1 + \frac{\sigma A L^2}{EI\lambda_n^2}} \tag{2}$$

where $\lambda_n$ is the modal factor, $E$ the Young's modulus, $L$ the length of the beam, $A$ the resonator cross-sectional area, $\rho$ the density, $\sigma$ the tensile stress, and $I$ the geometric momentum of inertia. In the case of DNA resonators with a circular cross section of radius $R$, the cross-sectional area and the momentum of inertia are defined according to:

$$A = \pi R^2; \quad I = \frac{\pi R^4}{4} \tag{3}$$

and insertion in Eq. (1) gives:

$$f_n = \frac{\lambda_n^2}{4\pi L^2} \sqrt{\frac{ER^2}{\rho}} \sqrt{1 + \frac{4\sigma L^2}{ER^2\lambda_n^2}} \tag{4}$$

From Eq. (3), resonance frequencies are found as the eigen-frequencies of the unstressed resonator, plus a term depending on the tensile stress. An approach to investigate the presence of tensile stress is the analysis on the modal factor, which depends on the boundary condition of the system under test. In the condition of the unstressed doubly clamped beam, the ratio between the resonance modes corresponds to $f_2/f_1 = 2.76$ and $f_3/f_2 = 1.96$. On the contrary, in a string, where tensile stress dominates and flexural rigidity can be neglected, the ratio tends to $f_2/f_1 = 2$ and $f_3/f_2 = 1.5$. Figure 6a shows the ratio of the resonance

frequency modes measured on pristine DNA and intercalated DNA with YOYO-1, GelRed, or CisPt. For all the four different types of DNA resonators, the ratio of the modes was found to be close to the values of bridges without stress. Supplementary Fig. 5a confirms that the four DNA resonators have values far from the mode ratio of a clamped–clamped beam dominated by stress. Hence, we can conclude that the suspension process based on the super-hydrophobic mechanism introduces a low tensile stress (which, in a first approximation, can be neglected) in the bundles.

A further confirmation of the beam configuration of the bundles, in which the flexural rigidity effect is dominating, comes from the analysis of the geometrical dependence of the resonance frequency. In the condition of the unstressed beam, the resonance frequencies of DNA bundles can be defined as:

$$f_n = \frac{\lambda_n^2}{4\pi} \frac{R}{L^2} \sqrt{\frac{E}{\rho}} \tag{5}$$

and thus, they are linearly dependent on the geometrical parameter $R/L^2$.

On the contrary, in a string configuration, when flexural rigidity can be neglected, the resonance frequencies become:

$$f_n = \frac{\lambda_n \pi}{L} \sqrt{\frac{\sigma}{\rho}} \tag{6}$$

where the modal factor assumes natural number values ($\lambda_n = 1, 2, 3\ldots$). Therefore, such resonance frequencies are linearly dependent on $1/L$. The frequencies of the fundamental mode measured on both pristine and intercalated DNA resonators clearly evidence a linear dependence on $R/L^2$, with a slope determined by the physical characteristics of the constitutive material (Fig. 6b). On the other hand, the plot of the resonance frequencies versus the inverse of the resonator length does not show any trend (Supplementary Figure 5b). Such analysis confirms that tensile stress in the DNA resonators used in this work is very small and that the resonance modes are mostly dependent on flexural rigidity. Therefore, from the measurement of the resonance frequencies of the bundles, using Eq. (5), it is possible to extract the value of the effective Young's modulus of the resonators. An effective Young's modulus considers that the organic resonators are composed of bundles of DNA strands and contain the effect of the Poisson's ratio, whose estimation is complex for this kind of composite nanostructures. The bundles composed of DNA intercalated with YOYO-1 and GelRed molecules present a higher effective Young's modulus (Fig. 6c). These bis-intercalants alter the pristine configuration of the nucleic acid double helix without any alteration of the Watson–Crick base pairing due to their mechanism of action; one molecule of the intercalator is inserted between adjacent base pairs, in the ratio 1:4. This variation severely modifies DNA mechanical properties: the intercalating molecules act as a structural reinforcement, stiffening the whole DNA string structure. The Young's modulus increases from $5.3 \pm 1.2$ GPa, as for the pristine DNA resonators, to $8.1 \pm 1.8$ GPa for the YOYO-1 sample and $12.6 \pm 1.6$ GPa for the GelRed one. On the contrary, the intercalation with CisPt lowers the Young's modulus to $2.6 \pm 1$ GPa, suggesting a softer DNA structure. In this case, CisPt covalently bounds preferentially to the N7 (nitrogen at position 7) of the guanine rings and causes the local double-helix deformation, decreasing the mechanical rigidity of the DNA bundle. The Young's modulus values are related to the elastic properties of the whole bundle and thus are comparatively larger than those reported in the literature for a single DNA strand[39–42], while they are comparable with the elastic modulus of DNA nanofibers, ranging from 1 to 15 GPa[43,44]. They give an indication of the

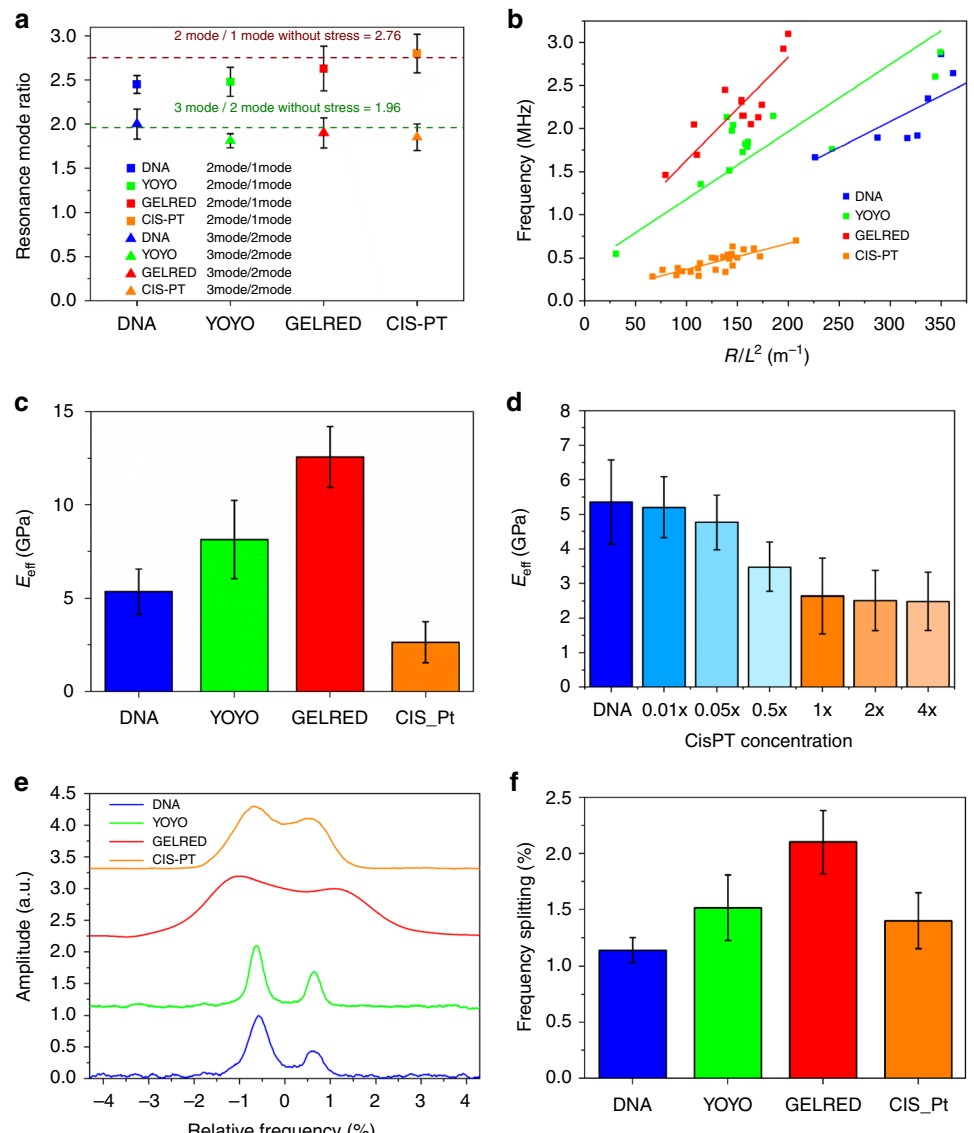

**Fig. 6** Mechanical properties of intercalated DNA. **a** Resonance mode ratio evaluated on bare DNA resonators and DNA intercalated with YOYO-1, GelRed, and CisPt. Dashed lines represent the theoretical value of the ratio between the resonance modes for a clamped–clamped beam without stress. **b** Frequency of the fundamental resonance mode as a function of the ratio of the bundle radius over the square of the length. The squares correspond to the experimental data and the lines to the linear fit. **c** Effective Young's modulus of the four different types of DNA resonators. Error bars were computed from the standard deviation of 25 measurements. **d** Effective Young's modulus of DNA bundles prepared with different concentrations of CisPt. The 1× concentration is the sample reported in the other panels. **e** Splitting of the first flexural vibration mode observed in vacuum, due to the asymmetry in the bundle section geometry. The spectra are centered with respect to the average of the two flexural modes. **f** Relative frequency splitting of the flexural vibration mode of the four different types of DNA resonators. Error bars were computed from the standard deviation of 10 measurements. Source data are provided as a Source Data file

variation of the stiffness of the DNA strand induced by the intercalating molecules. Moreover, the trend of the elastic properties of the DNA bundles measured with the vibrational analysis is in line with the values obtained with the simulation discussed above (Fig. 3c), confirming that the rigidity of the bundles is directly related to the mechanical properties of the DNA strands. This suggests that the inter-strand interaction in the bundle does not affect much the relative mechanical behavior of the intercalated DNA. This can be expected, because the interaction variation due to the intercalant is responsible for the covalent bonding changes (induced by the intercalant molecules between bases). This means that intercalants affect intra-strand interactions; instead, the inter-strand interaction between helices is weakly affected by intercalants and is mainly due to the

interaction of neutralized phosphate groups by sodium ions in the backbone of DNA. With respect to previous works, where the elastic properties were evaluated with a direct approach by stretching the DNA with an AFM or an optical tweezer[21,45], here the method is fast and does not require complex measurements, avoiding any perturbation induced by the pulling anchoring. The vibrometric technique benefits the high-quality factor and low damping obtained in air/vacuum. This technique, when used in a liquid phase, due to a strong damping, gives a signal-to-noise ratio much worse than the air/vacuum measurements. On the other hand, liquid nanomechanical measurements are not necessary, as they do not provide any further insight on the mechanical properties of the DNA bundle. For the sake of clarity and completeness, we point out that the intrinsic limitation of

this approach is not due to the technique itself, whose reproducibility, signal-to-noise ratio, and sensitivity are good enough, but to the massive availability of the super-hydrophobic devices. For future massive measurements and applications, it will be necessary for interested groups or laboratories that the super-hydrophobic devices become commercially available. A correspondence with the effective Young's modulus trend shown in Fig. 3c was found in the quality factor values at the resonance condition for the bare DNA resonators and the intercalated DNA ones (Supplementary Fig. 6). The $Q$ factor is higher in resonators intercalated with fluorescent molecules, YOYO-1, and GelRed, with respect to the bare DNA bundle, while it decreases in the intercalated CisPt resonator. Since the values of the $Q$ factor depend on the rigidity of the structure, these results follow the same trend observed for the Young's modulus.

The intercalation, through different molecular interactions between the intercalant and the DNA constituents, also affects the eccentricity of the DNA bundles and the splitting of the flexural mode into two flexural modes. The magnitude of the splitting (measurable only in vacuum) is directly proportional to the difference between the two axes. Mathematically, the structure has two different momentum of inertia defined according to the major ($d_{max}$) and minor ($d_{min}$) axis, causing the emerging of two resonance frequencies:

$$I_M = \frac{\pi d_{max}^3 d_{min}}{4}; \quad I_m = \frac{\pi d_{max} d_{min}^3}{4} \qquad (7)$$

$$f_{M,m} = \frac{\lambda_n^2}{2\pi L^2}\sqrt{\frac{EI_{M,m}}{\rho A}} \qquad (8)$$

From Fig. 6 e, f, it is clearly evident how intercalating molecules deform the pristine DNA structure, increasing the eccentricity of the bundle. Intercalation of YOYO-1 and CisPt caused a similar increase in the frequency splitting, with a variation of 32% and 21% with respect to the pristine DNA bundle splitting, respectively. This phenomenon is much larger for DNA bundles intercalated with GelRed (around 82%). The slight deformation of the almost cylindrical structure of the DNA double strand is a clear sign of the structural perturbation caused by the different intercalant molecules. This very small effect can be determined only with our nanomechanical analysis, since, at the state of the art, it is not possible to be detected with other characterization techniques. Together with the above-presented results, this peculiarity confirms the effectiveness of our analysis for structural investigation of DNA molecules.

## Discussion

The above analysis shows a strong dependence of the mechanical properties and vibrational behavior of the DNA bundle for different intercalants. A previous work[22] based on mechanical tweezers evaluated the mechanical properties of DNA bundles under X-ray damaging. A direct comparison with this work is difficult because the two preparation methods and the final tasks are different. On the contrary, there are similarities that regard the study of the variation of the mechanical properties induced by modification of the helicoidal structure and the periodicity of the base sequence. In our work, the specificity of molecular anchoring affects quantitatively the Young's modulus of the bundles and other several quantities related to the vibration spectrum in a clear and distinguishable way. This suggests that the suspended bundle can be viewed as a qualitative/quantitative sensor of all molecules that interact with the DNA and modifies its pristine structural properties. The mechanical properties of suspended DNA bundles change in a measurable way as a function of the type and mode the molecules interact with it, as also predicted at

molecular level by our computational analysis. HRTEM direct imaging suggests that the main molecular variations occurring in the double helices after platinum drugs intercalation are structural modifications and not related to double-strand breaks. Double-strand breaks would produce much shorter DNA fragments that cannot be suspended and imaged with this pillar–pillar interdistance, as confirmed by control experiments. The method proposed here is general and can be used to follow a wide category of interactions between molecules and DNA, including DNA and proteins interaction and their quantitative mass determination.

Considering titration applications, we performed SMD simulations with intercalated DNA complexes at intercalant concentration lower than the saturation condition. We created DNA systems with eight YOYO-1 molecules or three CisPt molecules. In agreement with the above-reported data, the trend has been maintained. As expected, the relevant $E_{SMD}$ values are closer to those estimated for the pristine DNA (1.98 ± 0.17 GPa for DNA/YOYO-1 and 1.23 ± 0.12 GPa for DNA/CisPt) than those calculated for the saturated complexes. Experimentally, CisPt titration analysis has been performed oversaturating (4 ×, 2 ×), saturating (1 ×), and under-saturating (0.5 ×, 0.05 ×, and 0.01 ×) the DNA. The intercalated DNA was used to evaluate the alteration of the mechanical properties of bundles with respect to pristine DNA. A clear trend of Young's modulus as a function of CisPt concentration has been observed and reported in Fig. 6d. Oversaturating concentrations of CisPt show a plateau in the Young's modulus, suggesting that above stoichiometric saturation, the mechanical properties of the bundle are not altered by the addition of a more chemotherapeutic compound. Lower CisPt concentrations show clearly that the Young's modulus value tends to that of the pristine DNA: the lower the concentration, the higher the Young's modulus.

These results prove the actual capability of the vibrometric technique to measure the intercalant molecular dosage effectively bound to the pristine DNA. The data presented in Fig. 6d suggest two possible future uses of the sensor: in vitro studies aiming at personalized chemotherapeutic administration and in environmental epidemiological studies of heavy metal effects on DNA of human/animal and plant origin. We notice that the experimental error in Fig. 6d can be further reduced. In future works, we will make an effort to improve the suspension process control conditions, such as temperature, humidity, and buffer composition, in order to obtain a large number of bundles with higher uniformity in size and composition.

Adequate medication time, rate, and dosage are fundamental tasks to ensure patients safety[46] and accuracy of the therapy[47]. Also, curative effects of chemotherapy compounds are frequently associated with undesired toxic consequences on healthy cells. The development of a tool that investigates the specificity and selectivity of a drug may shed light on clinical side effects. In this context, the direct study of DNA, extracted from cultured cells exposed to metal-based chemotherapeutic compounds, e.g., platinum-based substances, is greatly unexplored and the development of specific treatments based on in vitro long-term evaluations is of tremendous importance. We believe that our approach could prove to be useful also as a valuable diagnostic tool for evaluating intracellular damage induced by chemotherapeutic drugs on different types of cells. Moreover, moving from a cultured cell to patient-specific primary cultures could represent an important step forward in the development of personalized chemotherapeutic dosing.

The genome can be strongly affected also from heavy metals, considered among the most important environment contaminants. The exposure to such compounds, typically via contaminated water and food, leads to neurotoxicity, genotoxicity,

and carcinogenicity, due to the production of reactive oxygen species[48], to a major affinity to DNA if compared with proteins, to DNA damage[49,50], and to alterations to the double-strand break repair[51]. Some metals are characterized by the interaction with the DNA basis and/or backbone[52]; i.e., $Al^{3+}$ displaces metals such as $Mg^{2+}$ or $Fe^{3+}$ and, due to the greater affinity to DNA, inhibits enzymatic activity. In all the mentioned cases, the effects on the double helix are metal specific and dose dependent[51]. We believe that also in this case, our findings can be used as a method to verify the compound effects on DNA of cultured cells/primary cultures and to shed light on the alterations of pristine DNA and its functional conditions, thus contributing to answer a strongly debated question concerning the genomic variations occurring after exposure to heavy metals.

## Methods

**Super-hydrophobic device fabrication**. Super-hydrophobic samples were obtained as previously reported[24]. Si < 100 > was used as a substrate. Briefly, a circular array of pillars was defined by positive optical lithography, and metals were deposited by a sputter coater. The metals deposited were a 10 nm layer of Ti to promote adhesion of gold on Si, followed by a 50 nm Au layer, protected by 50 nm of Cr. Pillars of an approximate height of 10 μm were obtained by etching in a DRIE system and the Cr layer was removed with a selective wet etching. The sample was first covered with a 1-nm-thick layer of $Al_2O_3$, deposited by means of atomic layer deposition, and then functionalized with perfluorodecyltrichlorosilane (FDTS) in a Molecular Vapor Deposition System to make the device super-hydrophobic.

**DNA sample preparation and characterization**. λ-DNA (48502 bp sequence available at a producer's website, New England Biolabs, Ipswich, MA, USA) was preheated at 65 °C for 10 min before the incubation with a saturating amount of GelRed (Biotium, Hayward, CA, USA) or YOYO-1 (ThermoFisher, Waltham, MA, USA), MW: 1.271 g/mol, for 2 h at the constant temperature of 30 °C. DNA was diluted in saline buffer solution (6.5 mM NaCl, 10 mM Tris HCl, pH 9.3)[24] at the final concentration of 50 ng/μl. In the experiments with CisPt (Alfa Aesar, Karlsruhe, Germany), MW: 300 g/mol, λ-DNA was preheated at 65 °C for 10 min and then separately incubated with different amounts of the drug: saturating (1 ×), oversaturating (2 ×, 4 ×), and below saturation (0.5 ×, 0.05 ×, and 0.01 ×). The samples were incubated for 72 h in a water bath at 37 °C. A 5 μl droplet of the DNA or of the DNA/intercalant solution was then deposited by a pipette over a super-hydrophobic device. The evaporation occurred on a hot plate at the constant temperature of 25 °C[53], until the dewetting processes were completed. The samples were then imaged by SEM, working at an acceleration voltage of 5 kV and at 21 pA of current. In the cases of DNA intercalated with YOYO-1 and GelRed, the fluorescence of the suspended bundles was checked with a Nikon Eclipse $T_i$, equipped with a ×4 and a ×100 objective. YOYO-1 fluorescence was visualized by excitation in the band of 470/25 nm (center wavelength/FWHM) and emission in the range of 535/40 nm. GelRed fluorescence was visualized by excitation in the 540/25 nm band and emission in the 605/55 nm wavelength range. HRTEM imaging was performed by using a Titan 60–300 TEM working at 80 keV. The HRTEM was provided with a high-brilliance field-emission electron gun (X-FEG), a Wien-type monochromator, a spherical aberration corrector (CS) for the objective lens, and a Gatan Tridiem 865 image filter (GIF). The typical electron dose setting used for HRTEM imaging was about 10–20 electrons $\text{Å}^{-2}$ per second, and the exposure time of the CCD camera ranged from 0.3 to 1 s.

**Nanomechanical characterization**. Vibrational analysis of suspended DNA bundles was performed with a Laser Doppler Vibrometer system (MSA-500, Polytec Gmbh). The samples were mounted on a piezoelectric disk actuator inside a vacuum chamber. The laser (wavelength 633 nm) was focused on the DNA bundle with a ×100 objective to obtain a spot of around 1 μm in diameter. The measurements were done both in air and under vacuum. For the measurement in a vacuum environment, the chamber was evacuated with a pumping system composed of a series of turbomolecular and membrane pumps (MINI-Task System, Varian Inc. Vacuum Technologies) reaching a vacuum level around $10^{-7}$ mbar. The piezodisk was glued on a Peltier cell to perform the measurement at a fixed temperature (25 °C). The humidity of the chamber was controlled with a conditioning system and fixed at RH 60%. The measurements on the DNA resonator were performed at stable environmental conditions, at 25 °C and RH of 60%, to ensure full comparison with HRTEM data performed on A-form DNA. Relative humidity higher than 75% will cause a nucleic acid transition from the A- to the B-form, causing a structural variation, such as the interbase distances, tilt of the base pairs and diameter.

**Computational details**. The crystal structure [PDBid: 108d] of a DNA fragment in complex with a TOTO molecule was used as a base to develop our systems. Given

the great structural similarity between TOTO and YOYO-1 molecules, we fitted the YOYO-1 structure to the coordinates of the TOTO molecule in the crystal and then this fragment was repeated to obtain the final 66-bases long DNA filament with the following corresponding GCTAGCTGGCTAGCTGGCTAGCCAGCT AGCCAGCTGGCTAGCCAGCTAGCTGGCTAGCTAGCTAGC. The same sequence has been adopted to simulate the pristine DNA and the CisPt/DNA complex. All MD simulations were carried out with the AMBER 18 platform running on our CPU/GPU hybrid cluster. DNA was parameterized with AMBER OL15 forcefield[54], while parameters for the YOYO-1 molecule were derived with our standard procedure[54,55]. Parameters for guanosines bonded to cisplatin, and for cisplatin itself, were taken from the literature[56]. All simulations were carried out in implicit solvent, given the high box dimensions that would be needed for such elongated structures. After minimization, we performed a 100 ns long molecular dynamic simulation (integration time step of 1 fs) to get an equilibrated structure for each system, used as a starting point for the Young's modulus calculation. VMD software was used to visualize trajectories and produce images and videos.

**Steered molecular dynamics for Young's modulus calculation**. A dummy atom (DU) was placed on the vector joining the two edges of the DNA helices, at a distance corresponding to twice the value of the DNA length. The position of DU and of the two terminal phosphate groups at the opposite side of the DNA chain was restrained during all the simulations with a harmonic restrain constant of 20 kcal $\text{mol}^{-1}$ $\text{Å}^{-2}$. Eight different simulations were performed for each system, varying uniformly the applied force used to pull the free terminal phosphate groups toward DU. Each SMD simulation was performed three times, each lasting 20 ns, to stabilize the equilibrium length. The strain was then calculated by the following equation:

$$\text{Strain} = (\Delta L/L_0) \cdot 100 \qquad (9)$$

where $\Delta L$ is the length of the DNA under the effect of the pulling force, and $L_0$ is the starting DNA length. The stress was calculated as the force applied divided by the cross-sectional area of the helix, while the Young's modulus was derived from the slope of the stress–strain diagram after linear regression fitting.

## Data availability

All the data that support the findings of this study are available from the authors on reasonable request; see Author contributions for specific datasets. The raw data underlying Figs. 3b–d, 6a–f, and Supplementary Fig. 5a are provided as a Source Data file.

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

## Acknowledgements

The authors acknowledge financial support from King Abdullah University of Science and Technology for OCRF-2014-CRG and OCRF-2016-CRG grants and from Piedmont Region through European Funds for Regional Development ("Food Digital Monitoring" project).

## Author contributions

S.S. carried out the vibrational measurements. M.M. and M.A. prepared the samples. M.M., M.A., and S.L. performed the electron microscopy characterization. S.S. and C.R. analyzed and processed the nanomechanical data. D.M., E.L., and S.P. performed the molecular dynamics simulations. M.M. and E.D.F. conceived and analyzed the results from the structural, medical, and biological perspectives. S.S., M.M., C.F.P., C.R., and E.D.F. wrote the paper. All authors discussed the results and commented on the paper.

## Additional information

**Competing interests:** The authors declare no competing interests.

