## [Peer Review File · Nature Communications]

Reviewers' comments:

Reviewer #1 (Remarks to the Author):

The work presents interesting resonators fully composed of DNA forming bundles between microfabricated pillars. I am not aware of any other examples of resonators fully composed of a biological material and the relevance of the DNA molecule makes these resonators particularly interesting. The direct measurement of the vibration of the DNA-resonator by laser doppler vibrometry also proves to be an efficient approach to measure the most relevant mechanical characteristics of these resonators.

In addition to the obvious interest of this novel bio-device for the nanomechanics and bionanomechanics communities, I also find the results presented here can open up opportunities for research in molecular biology, biomedicine and biotechnology. The presented method has the potential to serve in the advancement of our knowledge in the interplay among structure and function of relevant biomolecules.

The authors present relevant proof of concept of the usefulness of the technique to study the biomechanics of DNA under the effect of different intercalating molecules.

I find this piece of work relevant and of high quality, deserving publication in Nature communications, but only after and if some major questions can be satisfactorily answered.

1. A wider discussion about the applicability to study DNA in liquids, under different pH, temperature and ionic strength is necessary. Radiation damage to DNA and its effect on DNA mechanics has also been studied previously by optical tweezers (Microsystems & Nanoengineering volume 2, Article number: 16062 (2016)), this prior art should be discussed.

2. How the authors can explain that the asymmetry parameter (as calculated from the peak separation in figure 5 e is larger for the molecule GELRED than CIS-Pt?. The CIS-Pt causes the helix denaturation and disruption (as stated in the manuscript), so larger asymmetry could be expected.

3. The effect of relative humidity in the elasticity of DNA thin films is a relevant issue under study for decades (Biopolymers 26, 1637–1665, 1987, Scientific Reports 7, 536 (2017)). Do the authors find a variation of the Young's modulus for the measurements performed in air and in vacuum?. They necessarily differ in the hydration of the DNA resonators. Discussion about the effect of relative humidity/hydration of the DNA bundles is needed. The ambient conditions during measurements needs to be clarified (temperature, RH,..). Have they run controls for each of the intercalation experiments?.

4. The discussion and Outlook section should be revised thoroughly. The usefulness of this technique for drug dosing is obscure to me. The authors should explain how the present study can translate into a dose guide in vivo.

5. How has been the sensitivity calculated?. Have the authors considered the noise in the system (Allan variation,..)?. The noise data should be shown in the main text or supplementary materials. Stating the lower limit of detection for the DNA resonators will also be informative to the readers.

6. The splitting of the resonance peaks and the capability to obtain the asymmetry parameter is a very appealing result with this method. Still, I find that resolution of the peaks GELRED and CisPt is very poor. Which is the error in the measurement of the asymmetry?.

7. How has been the error in the Young's modulus calculated?.

Minor issues:

The readability of the manuscript should be improved and typos corrected.

Figure 3 lacks the labels a) b) to ease the follow up of the message.

The DNA bundles in Figure 3 are barely visible.

Homogenize notation (resonance frequency is labelled as f and f_r , ...)

Reviewer #2 (Remarks to the Author):

In this manuscript, the authors constructed a micro/nano device so called "DNA resonator" by suspend double strand DNA on micro fabricated Si pillars to study the vibration property of the DNA bundles with/without intercalant molecules. The SEM and fluorescent microscope images assured the successful suspension of DNA bundles and certain dye intercalations. The DNA bundles crossing Si micro-pillars presented typical vibration spectrum both in air and in vacuum, and the flexural frequency shifted when intercalant molecules were introduced. The differences of split peaks of the first flexural frequency observed in vacuum provided further details of the mechanical property changes of DNA bundles due to molecule intercalation.

Based on the experimental facts, I believe that, as the authors proposed, this DNA resonator based technic can be developed into a sensitive characterization method to analyze DNA/ligand interactions. However, this method has its own limitations, and a few issues need to be further addressed:

1. Comparing to other DNA mechanical study methods (AFM, optical/magnetic tweezers) performed in solution, this DNA resonator is exposed in air or in vacuum which cannot preserve the in-situ formation of the B-form DNA nor to study any dynamic phenomenon. It would be better if the author could also acknowledge certain limitations before or after claiming all the advantages (Line 296-298).

2. To support the outlook direction that this method is suitable of being applied to chemotherapy titration (Line 30 & 347), the authors should (and could) provide more evidences from the current experiments. For example, titrating the intercalation dyes (besides of using saturated condition) and acquiring gradually shifted flexural frequencies due to accumulated distortion. This experiment is strongly recommended for fully consideration of acceptance.

3. The authors should be more cautious of making arbitrary claims like the small deformation change can hardly detected by other techniques (Line 320-322). A few examples here:
DOI:10.1038/s41598-017-07796-3; DOI:10.1093/nar/gkm529.

In summary, I would suggest the acceptance of this manuscript after a minor revision.

Reviewer #3 (Remarks to the Author):

In this manuscript, the authors describe the production of DNA bundles (30-100nm in diameter) suspended between pillars, and the optical interrogation of their resonant behavior both without and with intercalating molecules. From the resonant behavior, the authors make arguments regarding the mechanical and structural properties of the DNA bundles, and finally list a couple of potential applications of this technology. The measurements described in this manuscript are interesting and relevant to the field, but much of the interpretation is lacking and several logical jumps are made. I have two major concerns about this paper (specific comments below). The first is that the authors are inferring relationships between resonant behavior and single-molecule structure (and changes therein), while neglecting intermolecular behavior which likely has a profound influence on the mechanical properties of this system. These are not single molecule resonators, they are bundles. This significant gap renders much of the interpretation questionable.

My second concern is that the motivation for this work is not well laid out. There are some handwaving arguments regarding how this may be useful to optimize medication dosing or understand the effects of heavy metals, but these concepts (as described by this paper) don't appear to make much sense on closer inspection. In order for this manuscript to be publishable, major revisions are required.

The motivation for doing this work is unclear. The Introduction does not motivate the work, it simply describes some previous research that is related. There are some references to mechanical measurements of single DNA molecules, but there are several published manuscripts describing mechanical measurements of DNA nanofibers or bundles that ought to be included as well. The applications described at the end of the manuscript are a bit confusing; the dosing methodology the authors claim is the current state of the art is from a reference dated 1916. Depending on the medication and the disease to be treated, there are certainly more modern dosing approaches. The claim that a technique from over 100 years ago is the current state-of-the-art is questionable and not appropriate for comparison. How the results from this manuscript relate to dosing is quite unclear, particularly as medication must interact with and pass through a wide range of biological structures (and these interactions affect efficacy); medications are generally not applied directly to DNA and the delivery mechanism and vehicle are critical. Figure S5- the legend does not appear to correspond with the X axis labels. For example, the "GelRed" column has two data points, one red square and one orange square. The red square in the legend is labeled GelRed 2/1, but the orange square is labeled CisPT 2/1. I would think they should both be GelRed. The same problem appears for most of the other data points (the column in which they are plotted doesn't necessarily correspond with the dataset indicated by the legend).

-Why would the 3rd mode be below the noise for CisPt intercalation but not for the other intercalants? The signal-to-noise in 4C looks pretty good (for pure DNA, 3rd mode)

-The comparison between figures 5b and S6 is not convincing. The linear fit in 5B is not particularly good (there is no goodness of fit parameter described, so there is no way to quantitatively evaluate this). I could certainly draw a line to fit any of the data sets in S6 and the variation would not necessarily be any worse than the variation in 5B (it is, of course, hard to tell given the difference in scaling). As there are no attempts to fit lines in S6, there is no way to verify the argument that the linear fits to $R(L^{-2})$ are better than those to L^{-1}

Line 270- typo, "liner" should be "linear"

Line 234- typo, "clamped-clamper" should be "clamped-clamped"

The assumed eccentricity of the DNA bundles should be imaged using TEM. One could embed the bundles in epoxy, then take thin slices and image the cross-section. This would provide experimental evidence for the interpretation provided. Line 210 indicates the eccentricity is barely observable via EM images, but this reader was not able to observe it at all.

A major concern with this paper is that the results appear to conflate what the authors believe is going on at the single DNA molecule level with what they are measuring in DNA bundles ("fibers" with diameters [30-100nm] that suggest multiple molecules in thickness).

First, the authors should report the diameters of the bundles obtained, and indicate the level of diameter variation observed along a single bundle. In other words, if a bundle is 100nm in diameter, is it 100nm +/- 1% along the length of the bundle, or +/- 10%? As the geometry of the bundle is one of the critical links between resonant behavior and microstructure/mechanical properties, this aspect must be extremely well characterized and reported. The diameter distribution between bundles should also be reported. Given the SEM measurement approach, what is the error in the diameter measurement (or, what is the resolution of this measurement?)

Second, given that the bundles are multiple molecules in thickness, it is hard to understand why Figure 2 would show the periodicity as expected for a single molecule. As the authors expecting that the DNA molecules in the bundle are aligned in some crystalline form such that the atoms are in some lattice? Why would that be? Why are the TEM images in Figure 2 only indicating the surface/edge of the bundle, and is it expected that the same behavior exists throughout the entire cross section? Also, it would be helpful to report the various sizes of the intercalating molecules.

Overall, the authors talk about what may be going on with the DNA at the single molecule level and how the intercalants may change single-molecule behavior. However, mechanical behavior of the bundle will depend not only on single-molecule behavior, but on intermolecular interactions and molecular organization. This is not discussed anywhere in this manuscript and is a major gap

in interpretation.

How do the HRTEM images indicate that platinum drug intercalation does not indicate double strand breaks (line 372)? How are these imaging data evidence for this behavior or lack thereof at the single molecule level?

In summary, while the concept described in this manuscript is interesting, there are several flaws in interpretation and motivation that must be addressed before it would be appropriate for publication.

Dear Editor,

We give you here *point by point* answers to reviewers' comments, including new parts we have added to the manuscript (they appear in blue in the text) emerged from their questions. In particular, we added a Molecular Dynamics study to clarify the microscopic behavior of our experimental system.

We would like to thank you and all reviewers for their careful reading and precise inquiry. We believe that the manuscript now has improved a great deal.

Reviewers' comments:

Reviewer #1 (Remarks to the Author):

The work presents interesting resonators fully composed of DNA forming bundles between microfabricated pillars. I am not aware of any other examples of resonators fully composed of a biological material and the relevance of the DNA molecule makes these resonators particularly interesting. The direct measurement of the vibration of the DNA-resonator by laser Doppler vibrometry also proves to be an efficient approach to measure the most relevant mechanical characteristics of these resonators.

In addition to the obvious interest of this novel bio-device for the nano-mechanics and bio-nanomechanics communities, I also find the results presented here can open up opportunities for research in molecular biology, biomedicine and biotechnology. The presented method has the potential to serve in the advancement of our knowledge in the interplay among structure and function of relevant biomolecules.

The authors present relevant proof of concept of the usefulness of the technique to study the biomechanics of DNA under the effect of different intercalating molecules.

I find this piece of work relevant and of high quality, deserving publication in Nature communications, but only after and if some major questions can be satisfactorily answered.

Question 1: A wider discussion about the applicability to study DNA in liquids, under different pH, temperature and ionic strength is necessary. Radiation damage to DNA and its effect on DNA mechanics has also been studied previously by optical tweezers (Microsystems & Nanoengineering volume 2, Article number: 16062 (2016)), this prior art should be discussed.

Answer 1: Our method allows preparing DNA from different solution conditions, that means different pH, ionic strength, buffer composition etc. and we can expose the DNA even to radiation damage (not in this study). After we suspend the DNA bundle, the mechanical measurements are made in dry condition (both in air and/or in vacuum). Our experimental setup allows also damage study with the dried sample (not in this study). In principle, the mechanical study of the suspended DNA bundle in liquid phase can be done (not in this study), but it should be executed in a times lap in the range of tens of minutes because the DNA can detach from the pillar. In case of longer measurements, there is also the possibility to “weld” the DNA ends on Silicon pillars by using FIB Platinum deposition (we already did it. See picture here). We successfully tested this method but we didn't pursue these measurements because they are time and process consuming and because for our task, we could do all preparation variations in liquid phase, prior the suspension. It remains a possible configuration

available for future experiments, even if the amplitude of vibration of DNA bundles in liquid would be so damped to be extremely difficult to be detected. In any case here we were interested in studying the effect of intercalation and its experimental measurability. Regarding the paper suggested by the referee (we missed this publication and we thank the referee), the authors there used mechanical tweezers for detecting DNA radiation damage. We included this paper in the references because it is consistent with our findings, even if a direct comparison is difficult because the two preparation methods and the final tasks are different. Moreover they don't give any info (no TEM measurements) on the bundle structure.

The red circles in figure shows the DNA “welding” on silicon pillars by Pt deposited with FIB.

Question 2: How the authors can explain that the asymmetry parameter (as calculated from the peak separation in figure 5 e is larger for the molecule GELRED tan CIS-Pt?. The CIS-Pt causes the helix denaturation and disruption (as stated in the manuscript), so larger asymmetry could be expected.

Answer 2: In this work, the asymmetry measurements are proof of concept of the possibility to evaluate the deformation of the bundles with respect to the theoretical cylindrical shape. The results are remarkable because this very small effect can be determined only with our nanomechanical analysis, since, at the state of the art, it is not possible to be detected with other characterization techniques.

We observed that intercalating molecules deform the pristine DNA structure increasing the eccentricity of the bundle. This slight deformation is a clear sign of the structural perturbation caused by the intercalant molecules. Anyway, at the current point, it is very difficult to understand the role of each intercalant and the impact it can have on the DNA bundle, which is composed by more than one strand. Moreover Cis-Pt causes the helix local deformation, but this alteration does not mean that it deforms the cylindrical shape of the double strands more than the fluorescent intercalants. As well, the impact on the whole bundle depends on the configuration the different DNA double strands assume, that can also minimize the deformation effect. To conclude, the bundle asymmetry is influenced by three main effects: the inner deformation of the dsDNA due to intercalant, the hydration water molecules content, the residual salt contained in the bundle. To elucidate these details it will be necessary further studies that for the moment are beyond the main task of this paper.

Question 3: The effect of relative humidity in the elasticity of DNA thin films is a relevant issue under study for decades (Biopolymers 26, 1637–1665, 1987, Scientific Reports 7, 536 (2017)). Do the authors find a variation of the Young's modulus for the measurements performed in air and in vacuum?. They necessarily differ in the hydration of the DNA resonators. Discussion about the effect of relative humidity/hydration of the DNA bundles is needed. The ambient conditions during measurements needs to be clarified (temperature, RH,..). Have they run controls for each of the intercalation experiments?

Answer 3: We agree with the reviewer that the effective Young's modulus of DNA bundle could be dependent on hydration. Indeed, we checked our data and found that the frequency shift from vacuum to air environment could be hardly ascribable only to the added fluid mass. By the way, estimation of Young's modulus was made from measurements taken always in air environment, thus taking into account of the DNA hydration. On the contrary, vacuum condition was used just for the eccentricity evaluation. Thus, the possible variation of the Young's modulus between air and vacuum is not affecting our separate conclusions.

The measurements of DNA resonators were performed at fixed ambient conditions. The temperature was fixed at 25°C by a Peltier cell on which was glued the piezo-disk. The humidity of the chamber was controlled with a conditioning system and fixed at RH 60%. The description of the ambient conditions was added to the experimental section.

At RH lower than 75%, DNA in its A-form is dominant (J.D. Watson, F.H.C. Crick, Molecular structure of nucleic acids, Nature 171 (1953) 737–738; R.E. Franklin, R.G. Gosling, Molecular configuration in sodium thymonucleate, Nature 171 (1953) 740–741; M.H.F. Wilkins, A.R. Stokes, H.R. Wilson, Molecular structure of deoxypentose nucleic acids, Nature 171 (1953) 738–740). A conformational transition to the B-form can be appreciated only at a RH over 75%; the use of the suspended DNA system herein proposed as it is in such a hydrated environment will led to the detachment of the DNA bundles from the pillar tops, precluding effective measurements on the device. The description of the ambient condition and the considerations on the RH are reported in the revised manuscript.

Question 4: The discussion and Outlook section should be revised thoroughly. The usefulness of this technique for drug dosing is obscure to me. The authors should explain how the present study can translate into a dose guide in vivo.

Answer 4: The discussion and outlook section has been revised as suggested from the reviewer; the advantages of the use of our novel approach to monitor long terms drugs effect and DNA alteration of cultured and primary cells are explained in the present version of the manuscript.

Question 5: How has been the sensitivity calculated?. Have the authors considered the noise in the system (Allan variation,..)?. The noise data should be shown in the main text or supplementary materials. Stating the lower limit of detection for the DNA resonators will also be informative to the readers.

Answer 5: The theoretical sensitivity of the DNA resonators was computed from equation 1.1. Knowing the resonance frequency of the samples and their mass, we computed a range of theoretical sensitivity, since the dimensions of the bundles are

slightly variable (length and diameter). Regarding the noise, we performed different consecutive measurements of the thermal noise spectrum of the bundles and evaluated the resonance frequency. Each measurement takes up to 0.5 s. Then these measurements were used to compute the Allan deviation and defined the frequency noise and thus the lower limit of detection (LOD). The LOD data has been added to the manuscript, while details of the computation have been inserted in the SI.

Question 6: The splitting of the resonance peaks and the capability to obtain the asymmetry parameter is a very appealing result with this method. Still, I find that resolution of the peaks GELRED and CisPt is very poor. Which is the error in the measurement of the asymmetry?.

Answer 6: The resolution of the peaks in vacuum is much lower with respect of the measurement in air, because of the lower signal-to-noise ratio. This discrepancy is due to the small vibration introduced by the vacuum system that increase the noise in the measurement performed with the optical vibrometer. The error in the frequency splitting was computed from the standard deviation of more than 10 measurements of the splitting of DNA bundles for each different intercalant and for the bare double strand samples.

Question 7: How has been the error in the Young's modulus calculated?.

Answer 7: Similarly of the asymmetry measurement, the error in the Young's modulus evaluation was computed from the standard deviation of more than 20 measurements of DNA bundles in air for each kind of sample.

Minor issues:

Question 8: The readability of the manuscript should be improved and typos corrected. Figure 3 lacks the labels a) b) to ease the follow up of the message. The DNA bundles in Figure 3 are barely visible. Homogenize notation (resonance frequency is labelled as f and f_r , ...)

Answer 8: The paper was modified according to the minor issue highlighted by the reviewer.

Reviewer #2 (Remarks to the Author):

In this manuscript, the authors constructed a micro/nano device so called “DNA resonator” by suspend double strand DNA on micro fabricated Si pillars to study the vibration property of the DNA bundles with/without intercalant molecules. The SEM and fluorescent microscope images assured the successful suspension of DNA bundles and certain dye intercalations. The DNA bundles crossing Si micro-pillars presented typical vibration spectrum both in air and in vacuum, and the flexural frequency shifted when intercalant molecules were introduced. The differences of split peaks of the first flexural frequency observed in vacuum provided further details of the mechanical property changes of DNA bundles due to molecule intercalation.

Based on the experimental facts, I believe that, as the authors proposed, this DNA resonator based technic can be developed into a sensitive characterization method to analyze DNA/ligand interactions. However, this method has its own limitations, and a few issues need to be further addressed:

Question 1: Comparing to other DNA mechanical study methods (AFM, optical/magnetic tweezers) performed in solution, this DNA resonator is exposed in air or in vacuum which cannot preserve the in-situ formation of the B-form DNA nor to study any dynamic phenomenon. It would be better if the author could also acknowledge certain limitations before or after claiming all the advantages (Line 296-298).

Answer 1: As underlined by the reviewer, after the claim of the advantages of our technique, we introduce a comment regarding the limitation of our approach with respect of other method.

“On the counterpart, this technique presents some limitation related to the study in liquid environment of DNA properties and its dynamic interaction with other molecules.”

Question 2: To support the outlook direction that this method is suitable of being applied to chemotherapy titration (Line 30 & 347), the authors should (and could) provide more evidences from the current experiments. For example, titrating the intercalation dyes (besides of using saturated condition) and acquiring gradually shifted flexural frequencies due to accumulated distortion. This experiment is strongly recommended for fully consideration of acceptance.

Answer 2: We started this kind of experiments, but accurate and complete experimental titration is a long-term goal of our research project. Anyways, we added MD simulations of DNA behavior as a function of different concentration to demonstrate the sensitivity of the technique. Intercalants concentrations lower than those corresponding to the saturation conditions were considered; the results obtained showed a decrease of the corresponding effect on the Young's modulus values. The titration simulation has been commented and reported in this manuscript version.

Question 3: The authors should be more cautious of making arbitrary claims like the small deformation change can hardly be detected by other techniques (Line 320-322). A few examples here: DOI:10.1038/s41598-017-07796-3; DOI:10.1093/nar/gkm529.

Answer 3: We agree with the reviewer that small deformation of the cylindrical structure in principle can be measured also with other techniques. The difference between the two axes is between 0.3 and 1.5 nm, which are dimensions that can be evaluated with a good AFM or with TEM.

Instead experimentally, we claim the effect we are evaluating is not possible to be detected with other techniques. The eccentricity we are evaluating is an average effect over all the structure, which is around 12 μm long. Both AFM and TEM techniques would require too much time to evaluate all the variance on the bundle. In addition, our bundles are suspended over two pillars, while AFM approach would need the bundle to be placed on a substrate, which could introduce some artifacts or modify the measurement (slight deformation in the contact point). Instead, TEM would not provide the capability to evaluate the cross section of the bundles, but only a measurement of the diameter in one dimension (perpendicular to the electron beam axis).

The work of the references cited by the reviewer reported the measurements of very small variations of DNA double strand by AFM or optical techniques, but they are related to other kinds of phenomena with respect to ours. One work is related to the twisting of two separated DNA origami together, which is happening on a bigger scale (tens of nm). The other work is about the unwinding of ds-DNA because of intercalation. This work is very interesting since the authors are able to measure the unwinding of DNA by measuring the rotation of beads attached to the DNA strand. However, the phenomenon is different with respect to our system, because they are evaluating a dynamic effect by the rotation of the system, while we are measuring a static property of the sample (eccentricity of the cylindrical bundle), which is induced by the intercalation performed before the suspension of the bundle.

For these reasons, we are confident to claim that the DNA bundle eccentricity effect that we are evaluating is not possible to be detected with other techniques.

In summary, I would suggest the acceptance of this manuscript after a minor revision.

Reviewer #3 (Remarks to the Author):

Question 1: In this manuscript, the authors describe the production of DNA bundles (30-100nm in diameter) suspended between pillars, and the optical interrogation of their resonant behavior both without and with intercalating molecules. From the resonant behavior, the authors make arguments regarding the mechanical and structural properties of the DNA bundles, and finally list a couple of potential applications of this technology. The measurements described in this manuscript are interesting and relevant to the field, but much of the interpretation is lacking and several logical jumps are made. I have two major concerns about this paper (specific comments below). The first is that the authors are inferring relationships between resonant behavior and single-molecule structure (and changes therein), while neglecting intermolecular behavior which likely has a profound influence on the mechanical properties of this system. These are not single molecule resonators, they are bundles. This significant gap renders much of the interpretation questionable. My second concern is that the motivation for this work is not well laid out. There are some handwaving arguments regarding how this may be useful to optimize medication dosing or understand the effects of heavy metals, but these concepts (as described by this paper) don't appear to make much sense on closer inspection. In order for this manuscript to be publishable, major revisions are required.

Answer 1: In order to clarify the role of the single double strand respect to the bundle, now we included the molecular dynamic simulation where we calculated the Young's modulus of the single double strand. The results reported in the paper now show that the behavior of the Young's modulus for 2 different intercalants and pristine DNA (we couldn't simulate the GelRed, because the chemical formula is not released by the company and it is an industrial secret) is very close to the experimental one. This suggests that the inter strand interaction doesn't affect the relative behavior of the intercalated DNA. This can be expected, because the interaction variation due to intercalant is due to chemical bonding changes (induced by the intercalant molecules), instead the inter strand interaction is not affected and is much weaker due to neutralized phosphate groups by Sodium ions in the backbone of DNA.

Question 2: The motivation for doing this work is unclear. The Introduction does not motivate the work, it simply describes some previous research that is related.

Answer 2: We included a clear statement in the introduction where we state that: "The motivation of this work is to demonstrate the effect of intercalant molecules, of medical and clinical interest, on DNA structure through mechanical vibrometric measurements of suspended DNA bundles."

Question 3: There are some references to mechanical measurements of single DNA molecules, but there are several published manuscripts describing mechanical measurements of DNA nanofibers or bundles that ought to be included as well.

The applications described at the end of the manuscript are a bit confusing; the dosing methodology the authors claim is the current state of the art is from a reference dated 1916. Depending on the medication and the disease to be treated, there are certainly more modern dosing approaches. The claim that a technique from over 100 years ago is the current state-of-the-art is questionable and not appropriate for comparison. How the results from this manuscript relate to dosing is quite unclear, particularly as

medication must interact with and pass through a wide range of biological structures (and these interactions affect efficacy); medications are generally not applied directly to DNA and the delivery mechanism and vehicle are critical.

Answer 3: The discussion and outlooks section has been thoroughly reviewed and clarified in the present version of the manuscript

Regarding mechanical measurements we added two new references: “Direct Mechanical Measurements of the Elasticity of Single DNA Molecules by Using Magnetic Beads” by S.B. Smith *et al.*, science 1992, and “Twisting and stretching single DNA molecules” T. Strick *et al.*, 2000 progress in biophysics and molecular biology. They regard the study of mechanical stress of single DNA molecules. These studies, together with ref.39 and 40 of the manuscript, report Young’s modulus values for a single DNA molecule which are comparatively lower than the elastic properties of our bundles. Anyway, from the SMD results for single double strand helices we show that the behavior of the Young's modulus for 2 intercalants and pristine DNA is very close to the experimental bundle measurements. This suggests that the inter strand interaction in the bundle doesn’t affect much the relative mechanical behavior of the intercalated DNA. Indeed, the purpose of our work was related to give an indication of the variation of the stiffness of the DNA strand induced by the intercalating molecules. We couldn’t find any previous study of DNA bundle with our configuration with twisted and nested structure (see our reference “Direct imaging of DNA fibers: the visage of the double helix” F. Gentile *et al.*, NanoLetters 2012.) where mechanical properties were considered. In case, we would be happy to refer to papers that treat DNA bundles with similar configuration. Moreover, if the referee would kindly send us additional references we would be happy to comment and include them in the paper.

Question 4: Figure S5- the legend does not appear to correspond with the X axis labels. For example, the “GelRed” column has two data points, one red square and one orange square. The red square in the legend is labeled GelRed 2/1, but the orange square is labeled CisPt 2/1. I would think they should both be GelRed. The same problem appears for most of the other data points (the column in which they are plotted doesn’t necessarily correspond with the dataset indicated by the legend).

Answer 4: We thank you the reviewer for highlighting the formatting problem of Figure S5. The reference to the data point were uncorrected. We modify the figure accordingly.

Question 5: Why would the 3rd mode be below the noise for CisPt intercalation but not for the other intercalants? The signal-to-noise in 4C looks pretty good (for pure DNA, 3rd mode)

Answer 5: We performed additional measurements of the CisPt intercalated samples and were able to measure also the third resonance mode. The obtained data were in line with our statement of clamped-clamped beams in a regime dominated by flexural rigidity. The point related to the ratio 3 mode/2 mode was added both in figure 5a and figure S5.

Question 6: The comparison between figures 5b and S6 is not convincing. The linear fit in 5B is not particularly good (there is no goodness of fit parameter described, so

there is no way to quantitatively evaluate this). I could certainly draw a line to fit any of the data sets in S6 and the variation would not necessarily be any worse than the variation in 5B (it is, of course, hard to tell given the difference in scaling). As there are no attempts to fit lines in S6, there is no way to verify the argument that the linear fits to $R(L^{-2})$ are better than those to L^{-1}

Answer 6: We agree with the reviewer that the data presented in the first version of the manuscript were difficult to compare. Then, we fit with a linear trend also the data of Figure S6 (1/L) and reported for both the analysis the R^2 coefficients of the fit in Table S1. Even if for some fitting the R^2 is not very high, the values related to flexural rigidity are always higher with respect of the stress regime one. Therefore, all the analyses reported in the work confirmed that the resonator composed by pristine or intercalated DNA are in a regime dominated by flexural rigidity and that in first approximation the internal stress component can be neglected.

Question 7: Line 270- typo, “liner” should be “linear”; Line 234- typo, “clamped-clamper” should be “clamped-clamped”

Answer 7: The paper was modified according to typos highlighted by the reviewer.

Question 8: The assumed eccentricity of the DNA bundles should be imaged using TEM. One could embed the bundles in epoxy, then take thin slices and image the cross-section. This would provide experimental evidence for the interpretation provided. Line 210 indicates the eccentricity is barely observable via EM images, but this reader was not able to observe it at all.

Answer 8: The embedding of the bundles in epoxy resin is not compatible with the suspended biomaterial: the liquid state of the resin will remove the bundles from their site while pouring the materials. Also, the surface tension of the viscous resin deposited on the samples will give a strong contribution to the disruption of the suspended bundles. Furthermore the difference between the two diameters is way beyond the detection limit. Statement in Line 210 has been removed.

Question 9: A major concern with this paper is that the results appear to conflate what the authors believe is going on at the single DNA molecule level with what they are measuring in DNA bundles (“fibers” with diameters [30-100nm] that suggest multiple molecules in thickness).

First, the authors should report the diameters of the bundles obtained, and indicate the level of diameter variation observed along a single bundle. In other words, if a bundle is 100nm in diameter, is it 100nm +/- 1% along the length of the bundle, or +/- 10%? As the geometry of the bundle is one of the critical links between resonant behavior and microstructure/mechanical properties, this aspect must be extremely well characterized and reported. The diameter distribution between bundles should also be reported. Given the SEM measurement approach, what is the error in the diameter measurement (or, what is the resolution of this measurement?)

Answer 9: The data of the variability of the diameter over the whole bundles has been added to the manuscript and is in average around 7%. In this, we included salt and hydration shell variation. Details on the measurement approach have been included in the supporting information with some SEM images of the diameter measurements.

Regarding the evaluation of Young's modulus of each single bundle of bare DNA or intercalated one, the variability of the diameter was not taken into account, but only the average value. Indeed, the value of Young's modulus for each DNA sample family was extracted from the dispersion of the values obtained by measuring several different bundles. Then, the error was computed as standard deviation of the dispersion.

Question 10: Second, given that the bundles are multiple molecules in thickness, it is hard to understand why Figure 2 would show the periodicity as expected for a single molecule. As the authors expecting that the DNA molecules in the bundle are aligned in some crystalline form such that the atoms are in some lattice? Why would that be? Why are the TEM images in Figure 2 only indicating the surface/edge of the bundle, and is it expected that the same behavior exists throughout the entire cross section?

Answer 10: A-DNA superstructures have been demonstrated by simulation in the paper "Gentile F. et al., Direct Imaging of DNA Fibers: The Visage of Double Helix, *NanoLetters* 2012 12 (12), 6453-6458" where a bundle constituted by 7 DNA helices (a central helix surrounded by a shell of 6 helices) has been simulated to obtain the correspondent TEM image. Molecular dynamics simulations on dsDNA aggregation free energy as a function of helices displacement in the z-direction also confirmed the ordered fiber formation: the minimum interaction energy has been obtained for dsDNA helices with a solvation shell of 0.3 nm and aligned.

The edge of the bundle is the outer shell of the fiber and therefore is not subject to the overlap with the other helices features, giving an HRTEM imaging free of any interference (refer to Figure 3B in Gentile *et al.*, *NanoLetters* 2012 12 (12), 6453-6458). In the inner part of the bundle, the superimposition of the backbone with the bases and of the bases (belonging to overlapping helices) with different base pair and base step parameters originate the complex behavior observed at HRTEM.

Question 11: Also, it would be helpful to report the various sizes of the intercalating molecules.

Answer 11: Molecular weights of the intercalants used have been reported in Materials and Methods section.

Question 12: Overall, the authors talk about what may be going on with the DNA at the single molecule level and how the intercalants may change single-molecule behavior. However, mechanical behavior of the bundle will depend not only on single-molecule behavior, but on intermolecular interactions and molecular organization. This is not discussed anywhere in this manuscript and is a major gap in interpretation.

Answer 12: In order to clarify the role of the single double strand respect to the bundle, now we included the molecular dynamic simulation where we calculated the Young's modulus of the single double strand. The results reported in the paper now, shows that the behavior of the Young's modulus for 2 different intercalants and pristine DNA (we couldn't simulate the GelRed, because the chemical formula is not released by the company and is an industrial secret) is very close to the experimental one. This suggests that the inter strand interaction doesn't affect the relative behavior of the intercalated DNA. This can be expected, because the interaction variation due

to intercalant is due to chemical bonding changes (induced by the intercalant molecules), instead the inter strand interaction is much weaker due to neutralized phosphate groups by Sodium ions in the backbone of DNA. We added a new text at the end of paragraph “DNA molecular dynamics simulation”.

Notice also that we produced suspended single DNA filament but its mechanical stability, under TEM and laser beams, is poor and it is difficult to have reproducible measurements for our mechanical and structural study.

Question 13: How do the HRTEM images indicate that platinum drug intercalation does not indicate double strand breaks (line 372)? How are these imaging data evidence for this behavior or lack thereof at the single molecule level?

Answer 13: The presence of double strand breaks is expected to originate short lambda DNA fragments that are not suitable to cover the pillar-pillar inter-distance; also, the formation of a bundle of 12 μm with such short fragments is highly disadvantaged.

In summary, while the concept described in this manuscript is interesting, there are several flaws in interpretation and motivation that must be addressed before it would be appropriate for publication.

Reviewers' comments:

Reviewer #1 (Remarks to the Author):

After the changes made and the answers received, I consider this work is suitable to be published in Nature communications.

Reviewer #2 (Remarks to the Author):

In the revised manuscript, improvements have been made, however the hard core concerns about the significance of this mechanical DNA resonator technique has not been well addressed. The simulation of the gradually twisted DNA double helix due to titrated intercalated dyes (Figure 3) presents itself well, but it doesn't support the article strong enough. Because, the simulation is based on double strand DNA instead of DNA bundles as used in the work, and the theoretical simulation of dsDNA cannot prove that the DNA bundle deformation under unsaturated intercalation status can be well detected/quantified by the resonator setup. Moreover, the selling point of this manuscript is the experimentally applicable technique instead of the theoretical DNA twisting exhibition.

On the other hand, a published work DOI:10.1038/s41598-017-07796-3 had presented a successful example of experimentally quantifying the twisting of DNA bundles (origami bundle: a structurally well-defined bundle instead of suspended bundle) according to the amount of intercalating dyes in micro molar scale in solution. However, this work as a comparable approach was not well appreciated in the author's reply. From a peer reviewer's point of view, I would consider the work in the Ref. mentioned above, using DNA origami rods system, a better candidate for chemotherapy drug titration application.

In summary, I insist to say that the experimental result of intercalating dye titration is a must for the consideration of publish in Nature Communication.

Reviewer #3 (Remarks to the Author):

I do not feel the authors adequately responded to many of the reviewer comments and concerns. Please see below:

Reviewer 1:

Q1: Requests discussion of applicability of the measurement technique in different environments. Authors state that the DNA may be prepared in different environments but do not discuss use of the technique in different environments, just air and vacuum. It does not appear that the request for additional discussion of prior art related to studying radiation damage was addressed, the authors simply added a reference without any discussion.

Q2: Reviewer asks for explanation as to why the asymmetry parameter for GELRED is larger than that for CIS-Pt. Authors say asymmetry parameter is due to many things and offer no single explanation for the issue raised by the reviewer. A more in depth discussion in the manuscript is warranted, even if the results cannot be fully explained.

Q3: Reviewer asks for discussion of the effects of ambient humidity on the measurements taken in air. No discussion of this was added to the manuscript except to state the ambient humidity at which the measurements were performed, and the state of the DNA at humidity below 75%. The fact that the humidity was controlled to 60% in the chamber suggests the comparisons between experiments are relevant and appropriate, but a discussion of the effects of humidity is still lacking.

Q4: Reviewer asks for clarity on how this technique could be used for dosing. Authors add additional discussion on this, but do not provide any clear methodology for one might interpret results from DNA extracted from patients and measured using their technique. It is clear that one COULD measure patient DNA after exposure for various drugs, but how the resonant/mechanical

properties revealed by this measurement approach relate to dosing and clinical efficacy is quite unclear. In other words, how can one use DNA bundle resonant frequency, calculated modulus, or asymmetry to determine dosing? This is the question that must be answered for this approach to make sense.

Q5: Addressed

Q6: The reviewer asked about the error in resolving the split peaks due to asymmetry. The answer is confusing; it looks like the peaks in vacuum are resolved much better than in air as they are higher Q, and even though there is some amplitude noise at the base of the peak, that doesn't seem to be affecting the signal at the peak itself. The fact that the error bars in Fig 5f are due to the standard deviation of 10+ experiments should be stated in the manuscript. Are the peaks well-enough resolved such that this error is due to measurement statistics and not low Q of the peaks? The Q of the split peaks in the top two plots in Fig 5e seems rather low.

Q7: Error bars being due to standard deviation of 20+ measurements should be stated in the manuscript, not just in the response to reviewers (Fig 5c)

Q8: Addressed

Reviewer 2:

Q1: Reviewer asks for discussions of the limitations of this approach, which is quite fair. Authors add a single sentence that is rather understated and vague. A more in-depth discussion of the limitations should be provided.

Q2: Relates to Reviewer 1 Q4: how can this approach be used for dosing? It appears that titration was performed in the newly-added simulations, and the simulation results shifted, but no experimental titration was performed, and no concept for how to interpret the results of a titration experiment are provided. This reviewer comment appeared to strongly suggest experimental titration experiments, but these were not done.

Q3: The reviewer suggests the eccentricity of the bundles could be measured by other techniques, but the authors object. The authors are correct that the effect they are observing is an average over the entire bundle, which a single AFM or TEM image would not provide, but in TEM the fiber could be imaged from multiple angles (i.e. rotate the bundle around its axis and take pictures every 5-10 degrees) and this measurement could be performed without too much trouble. The cross section could be imaged by embedding the bundle in a matrix and then sectioning. They are correct that AFM imaging this eccentricity with AFM would be tricky, unless a portion of the bundle were somehow mounted vertically on a substrate (quite tricky, but not impossible). In any case, I think one could find a way to measure the eccentricity of the bundles another way.

Reviewer 3

Q1: Addressed

Q2: Somewhat addressed. The motivation from a pure knowledge perspective is now more clear, but from an application perspective is still somewhat murky. See Reviewer 2, Q2 and Reviewer 1, Q4

Q3: Please see the following references:

doi.org/10.1002/mame.201800302

doi.org/10.1039/B717581G

doi.org/10.1116/1.2801886

doi.org/10.1021/nl203503s (has some good references in the DNA origami area)

Q4: Addressed

Q5: Addressed

Q6: The addition of R^2 values helps tremendously. Addressed, except perhaps for CisPT...the

R² values here are not so different, why is that?

Q7: Addressed

Q8: While the proposed experiment is admittedly not trivial, it should be possible to image the bundle cross section. Low viscosity embedding resins exist, and even if the bundle were detached during embedding, would that mean that the eccentricity would be lost? The authors themselves, by frequency analysis, suggest that the tensile stress in the suspended bundles is quite small, so it isn't clear that detaching would be a major issue. The only issue that may admittedly be impossible to overcome perfectly would be ensuring the section is perfectly perpendicular to the bundle axis; this could render the sectioning approach meaningless. In any case, some secondary verification that the peak splitting is due to eccentricity would make this claim stronger. This reviewer agrees that eccentricity is certainly the most likely explanation, but a visual verification would be nice, particularly as it was given previously but withdrawn.

Q9: Addressed

Q10: Addressed

Q11: Addressed

Q12: The authors are now saying that the bundles are not stable under either TEM or laser...but aren't they using a laser to measure them? And a TEM to image them? What are the issues with respect to stability under a laser, and does that affect the results in this manuscript?

Q13: This answer suggests that the authors believe that, in order for a bundle to be formed spanning two pillars, the length of the DNA molecules composing the bundle must be at least the pillar-pillar distance...do the authors have proof for this? Are they suggesting each molecule starts and stops on a pillar? What if only 50% of the molecules were broken?

A minor comment:

1) The plots at the bottom left of the movies are impossible to read, the font for the axes labels and numbers is too small and so it isn't clear from watching the movie what the plot is meant to indicate.

Several of the initial issues with this manuscript were addressed, yet there remain some lingering issues that are pretty substantial. The one which all 3 reviewers seemed to have concerns about is how this technique would be used in the applications described; I do not think the authors have clearly stated this, and the simulated titrations do not do much to convince the reader of the utility. While the argument that bundle eccentricity causes peak splitting is reasonable and likely true, it would be more convincing if some additional evidence were provided for this...particularly since the image supposedly showing this evidence was removed because it didn't really do so.

Reviewers' comments:

Reviewer #1 (Remarks to the Author):

After the changes made and the answers received, I consider this work is suitable to be published in Nature communications.

A. We thank the reviewer for his positive comment.

Reviewer #2 (Remarks to the Author):

In the revised manuscript, improvements have been made, however the hard core concerns about the significance of this mechanical DNA resonator technique has not been well addressed.

The simulation of the gradually twisted DNA double helix due to titrated intercalated dyes (Figure 3) presents itself well, but it doesn't support the article strong enough. Because, the simulation is based on double strand DNA instead of DNA bundles as used in the work, and the theoretical simulation of dsDNA cannot prove that the DNA bundle deformation under unsaturated intercalation status can be well detected/quantified by the resonator setup. Moreover, the selling point of this manuscript is the experimentally applicable technique instead of the theoretical DNA twisting exhibition.

A We added the single double helix simulation because it reproduces qualitatively the trend behavior of the bundle. This is because we expect that the intercalant modification affects the intra-helix interactions more than the bundle inter-helix interaction. Nevertheless, we agree with the referee that a quantitative simulation (not available soon) would complete the explanation. As suggested by the reviewers, we decided to widen the paper and the experimental part has been strengthened and expanded to further support the results, whose interpretation is in agreement with the theoretical simulation already presented.

Q1: On the other hand, a published work DOI:10.1038/s41598-017-07796-3 had presented a successful example of experimentally quantifying the twisting of DNA bundles (origami bundle: a structurally well-defined bundle instead of suspended bundle) according to the amount of intercalating dyes in micro molar scale in solution. However, this work as a comparable approach was not well appreciated in the author's reply. From a peer reviewer's point of view, I would consider the work in the Ref. mentioned above, using DNA origami rods system, a better candidate for chemotherapy drug titration application.

A1: We thank the reviewer for the reference suggestion. The reference has been integrated in the new version of the manuscript and commented in the phrase "*Intercalator affects the DNA-superstructures and its mechanical properties. The study of self-assembled objects such as DNA-Origami have been previously reported [Direct Mechanical Measurements Reveal the Material Properties of Three-*

*Dimensional DNA Origami. Dominik J. Kauert, Thomas Kurth, Tim Liedl, and Ralf Seidel; Nano Lett. 2011, 11, 5558–5563, Twisting of DNA Origami from Intercalators. Reza M. Zadegan, Elias G. Lindau, William P. Klein, Christopher Green, Elton Graunard, Bernard Yurke, Wan Kuang & William L. Hughes. Scientific Reports 2011, 7: 7382]. Herein we are interested in achieving chemotherapeutics titration on DNA helices, autonomously self assembled without extensive preliminary thermal treatment or in-silico design requirements." The work suggested uses 6 helices bundles obtained by the hybridization of a scaffold with tens of staple strands. This approach may also be considered for chemotherapy drug titration. In our approach there is no *in-silico* design of any sequence and the bundle formation is autonomously driven by the droplet dehydration event and does not require linking points to keep the structure stable in dry condition on the pillars array. This guarantees the application of the suspension approach on genomic DNA, whose mechanism is different from DNA-origami technique (ref. The structure of DNA by direct imaging and related topics. Monica Marini, Tania Limongi, Manola Moretti, Luca Tirinato, Enzo di Fabrizio (2017). *La Rivista del Nuovo Cimento*, 5: 241-278.). This opens the way to the evaluation of the effects of chemotherapeutic drugs directly on human DNA expanding the field of applicability of the technique. This represents an element of novelty and certainly an advantage with respect to what is reported in literature so far.*

Q2: In summary, I insist to say that the experimental result of intercalating dye titration is a must for the consideration of publish in Nature Communication.

A2: We thank the referee for this stimulating request and we modified Figure 5d to show a complete CisPt titration experiment. The previous figure 5d related to the Q factor has been moved to the Supporting Information. The titration experiment has been commented in the "DISCUSSION AND OUTLOOKS": "*CisPt titration experiments have been performed oversaturating (4x, 2x), saturating (1x) and non-saturating (0.5x, 0.05x, 0.01x) the DNA. The intercalated DNA was used to evaluate the alteration of bundles mechanical properties with respect to pristine DNA. A clear trend of Young's modulus as a function of CisPt concentration has been observed and reported in Figure 5d.*

Oversaturating concentrations of CisPt show a plateau in the Young's modulus, suggesting that above stoichiometric saturation the mechanical properties of the bundle are not altered by the addition of more chemotherapeutic compound. Lower CisPt concentrations show clearly that the Young's modulus value tends to that of the pristine DNA: the lower the concentration, the higher the Young's modulus.

These results prove the actual capability of the vibrometric technique to measure the intercalant molecular dosage effectively bound to the pristine DNA. The data presented in Fig. 5d suggests two possible future uses of the sensor: in in-vitro studies aiming at personalized chemotherapeutic administration and in environmental epidemiological studies of heavy metal effects on DNA of human/animal and plant origin. "

Reviewer #3 (Remarks to the Author):

I do not feel the authors adequately responded to many of the reviewer comments and concerns. Please see below:

Reviewer 1:

Q1: Requests discussion of applicability of the measurement technique in different environments. Authors state that the DNA may be prepared in different environments but do not discuss use of the technique in different environments, just air and vacuum. It does not appear that the request for additional discussion of prior art related to studying radiation damage was addressed, the authors simply added a reference without any discussion.

A1: We thank the referee to give a chance to better explain our findings. In the present approach we used a physiologically compatible biological preparation procedure. The DNA/intercalator recognition event occurs in a liquid environment and the variation of parameters such as buffer pH, ionic strength, salts and concentrations can be fully customized prior to the deposition over the super hydrophobic device. The dehydration after the suspension of the bundles between micro-pillars does not affect the chemical reactions occurred in solution and the related modifications nor the pristine DNA condition (*e.g.* there is no overstretching, understretching, denaturation etc).

The measurement in air and vacuum were necessary for our vibrometric and high-resolution studies and imaging on DNA intercalation. The proof of concept of the vibrometric technique benefits of the high quality factor and low damping obtained in air/vacuum. The vibrometric technique used in a liquid phase, due to strong damping, gives a signal to noise ratio much worse than the air/ vacuum measurements. On the other hand, liquid vibrometric measurements are not necessary, as they would not provide any further insight on the mechanical properties of the DNA-bundle.

This limitation is underlined in the manuscript at the end of the Results section with the comment *The vibrometric technique benefits of the high quality factor and low damping obtained in air/vacuum. This technique, when used in a liquid phase, due to strong damping, gives a signal-to-noise ratio much worse than the air/ vacuum measurements. On the other hand, liquid nanomechanical measurements are not necessary, as they do not provide any further insight on the mechanical properties of the DNA-bundle.*

The comment on the damping has been added in the current version of the manuscript.

Regarding the paper suggested by the referee, we included this paper in the references to further argument and support our findings. We inserted a comment in the DISCUSSION AND OUTLOOKS section *"Previous work²⁷ based on mechanical tweezers evaluated the mechanical properties of DNA bundles under X-ray damaging. A direct comparison with the present work is difficult because the two preparation methods and the final tasks are different. On the contrary, there are similarities that regard the study of the variation of the mechanical properties induced by modification of the helicoidal structure and the periodicity of the base sequence. In our work the variation is related to the presence of intercalants, while in the related paper it is due to the X-ray radiation."* We inserted also a comment in the Introduction related to this part: *"It was demonstrated that the degradation of the*

periodic DNA structure under therapeutic X-ray radiation causes a noteworthy worsening of its mechanical properties²⁷”

Q2: Reviewer asks for explanation as to why the asymmetry parameter for GELRED is larger than that for CIS-Pt. Authors say asymmetry parameter is due to many things and offer no single explanation for the issue raised by the reviewer. A more in depth discussion in the manuscript is warranted, even if the results cannot be fully explained.

A2: As we stated in the answer to the Reviewer 1 in the previous revision, the eccentricity measurement is an estimation of the deformation of the bundles with respect to the theoretical cylindrical shape. It was observed that intercalating molecules deform the pristine DNA structure increasing the eccentricity of the bundle, but the detailed mechanism of this distortion and the role of each intercalant needs to be further investigated. We suppose that bundle asymmetry is influenced by three main effects: the inner deformation of the dsDNA due to intercalant, the residual water molecules content and the residual salt contained in the bundle. In any case, the main core of the paper refers to the stiffness changes, while we only pointed out that the vibrometric measurements allow to detect this very small bundle deformation (the deformation ranges from 0.35 to 1.15 nm that is only about 1 % with respect of the circular shape); obtaining a similar information would be extremely difficult or impossible with advanced direct imaging such as the one provided by HRTEM (see answer question #3 below).

Q3: Reviewer asks for discussion of the effects of ambient humidity on the measurements taken in air. No discussion of this was added to the manuscript except to state the ambient humidity at which the measurements were performed, and the state of the DNA at humidity below 75%. The fact that the humidity was controlled to 60% in the chamber suggests the comparisons between experiments are relevant and appropriate, but a discussion of the effects of humidity is still lacking.

A3: Surely humidity can impact the measurement, that's why we carefully control it. But again, the molecular recognition is made in the liquid phase, while the best controlled ambient conditions (Vacuum or air at RH and T fixed) are employed for the vibrometric measurement. We included in the manuscript the comment in the paragraph "Nanomechanical DNA resonator analysis": *"The measurements on the DNA resonator were performed at stable environmental conditions, at 25°C and RH of 60%, to ensure full comparison with HRTEM data performed on A-form DNA. Relative humidity higher than 75% will cause a nucleic acid transition from the A- to the B-form, causing a structural variation such as the interbase distances, tilt of the base pairs and diameter. This study is focused on the intercalation structural changes due to reactions already occurred in liquid phase, and then analyzed in standard measurement conditions without undergoing further irrelevant (for the present study) structural transition due to overcoming humidity threshold of 75%."*

Q4: Reviewer asks for clarity on how this technique could be used for dosing. Authors add additional discussion on this, but do not provide any clear methodology

for one might interpret results from DNA extracted from patients and measured using their technique. It is clear that one COULD measure patient DNA after exposure for various drugs, but how the resonant/mechanical properties revealed by this measurement approach relate to dosing and clinical efficacy is quite unclear. In other words, how can one use DNA bundle resonant frequency, calculated modulus, or asymmetry to determine dosing? This is the question that must be answered for this approach to make sense.

A4: As for Referee 2, we thank the referee for this request that stimulated a complete experiment to demonstrate how the CisPt titration can be followed through the Young's modulus variation. Now we have added results on the mechanical properties as a function of CisPt concentration.

The results on the additional measurements are reported in Fig. 5d and the related discussion is reported in the paragraph "DISCUSSION AND OUTLOOKS". Additional information about the titration parameters is reported in the "Materials and Methods" section. Such results prove the actual capability of the vibrometric technique to measure the intercalant molecular dosage effectively bound to the pristine DNA.

Q6: The reviewer asked about the error in resolving the split peaks due to asymmetry. The answer is confusing; it looks like the peaks in vacuum are resolved much better than in air as they are higher Q, and even though there is some amplitude noise at the base of the peak, that doesn't seem to be affecting the signal at the peak itself. The fact that the error bars in Fig 5f are due to the standard deviation of 10+ experiments should be stated in the manuscript. Are the peaks well-enough resolved such that this error is due to measurement statistics and not low Q of the peaks? The Q of the split peaks in the top two plots in Fig 5e seems rather low.

A6: The statement of the standard deviation was added in the caption of figure 5. The Q factor of the vacuum peak is higher compared to the air measurements and this fact allowed appreciating the splitting, which is not observable in the air measurements. For the two top measurements, the Q factor is lower than the others and therefore there is a partial overlapping of the peaks. Anyway, they were well-enough resolved to be detected singularly and the error in Fig. 5f comes from the measurement statistics.

Q7: Error bars being due to standard deviation of 20+ measurements should be stated in the manuscript, not just in the response to reviewers (Fig 5c)

A7: The statement was added in the caption of Figure 5.

Reviewer 2:

Q1: Reviewer asks for discussions of the limitations of this approach, which is quite fair. Authors add a single sentence that is rather understated and vague. A more in-depth discussion of the limitations should be provided.

A1: We added the comment related to the limitations of our approach in the "Effect of the intercalant on the DNA mechanical properties." section.

" For the sake of clarity and completeness, we point out that the intrinsic limitation of the present approach is not due to the technique itself whose reproducibility, signal-to-noise ratio and sensitivity are good enough, but to massive availability of super-hydrophobic devices. For future massive measurements and applications, it will be necessary for interested groups or laboratories that the super-hydrophobic devices become more reliable and commercially available. The availability of a big number of devices, routinely fabricated, will improve the DNA sampling and the statistical error in the Young modulus determination. "

Q2: Relates to Reviewer 1 Q4: how can this approach be used for dosing? It appears that titration was performed in the newly-added simulations, and the simulation results shifted, but no experimental titration was performed, and no concept for how to interpret the results of a titration experiment are provided. This reviewer comment appeared to strongly suggest experimental titration experiments, but these were not done.

A2: We added the titration experiments. Comments to the newly added experiments are reported in the answer 2 to the Reviewer #2 and answer 4 of the Reviewer #3.

Q3: The reviewer suggests the eccentricity of the bundles could be measured by other techniques, but the authors object. The authors are correct that the effect they are observing is an average over the entire bundle, which a single AFM or TEM image would not provide, but in TEM the fiber could be imaged from multiple angles (i.e. rotate the bundle around its axis and take pictures every 5-10 degrees) and this measurement could be performed without too much trouble. The cross section could be imaged by embedding the bundle in a matrix and then sectioning. They are correct that AFM imaging this eccentricity with AFM would be tricky, unless a portion of the bundle were somehow mounted vertically on a substrate (quite tricky, but not impossible). In any case, I think one could find a way to measure the eccentricity of the bundles another way.

A3: The above-suggested techniques have some characteristics which limit the applicability to our device in the present configuration:

1- The eccentricity evaluated relies on the average effect on 12 μm bundle length, and this unfortunately imposes a limitation on the use of HRTEM. In our HRTEM metrological measurements we performed a statistic on 3 images for each type of sample (pristine DNA, DNA/YOYO-1, DNA/GelRed, DNA/CisPt), measuring the bundle diameter on a sample length of 150 nm every 25 nm. The images used are related to a bundle placed orthogonal to the HRTEM e-beam working in optimized condition. An example of HRTEM micrograph with the histogram related to diameter size is reported in the picture below.

We found a fluctuation of the relative diameter in the range of approximately the 3-5%, already higher than the eccentricity variation percentage of 1.15% as from vibrational analysis. The bundles diameter analysis is reported in the Table below. The acquisition of tilted images, as we performed several times in the past, would show a comparable fluctuation in the diameter. The averaged variation that is sensed by our vibrometric method is beyond the detection capability of the HRTEM, which only gives local information on the diameter change, but not such a precise

information on the average difference. When tilting the sample, the HRTEM on the bundle would only produce local information that would not be of any further insight. Moreover, since the TEM measurement fluctuation is already on a local scale much larger than the difference we can appreciate with the vibrometer, also a statistics on the same bundle in consecutive frames taken at different position would not result in an improvement of the sensitivity.

Figure. HRTEM image of a DNA/YOYO-1 bundle, placed orthogonal to the e-beam. In the inset histogram related to the bundle diameter size.

	CisPt			GelRed		
	Image 1	Image 2	Image 3	Image 1	Image 2	Image 3
	19.81	18.18	34.97	28.06	30.08	23.94
	18.94	18.75	34.04	29.06	30.39	24.06
	18.06	20.33	32.01	27.98	28.54	24.62
	19.66	20.22	32.58	30.75	30.21	25.91
	20.75	19.88	30.66	29.75	28.98	24.73
STD	1.01	0.96	1.69	1.17	0.83	0.78
Mean	19.44	19.47	32.85	29.12	29.64	24.65
STD/Mean %	5.18%	4.91%	5.16%	4.02%	2.79%	3.17%

	YOYO-1			DNA		
	Image 1	Image 2	Image 3	Image 1	Image 2	Image 3
	28.63	9.78	26.91	24.10	27.55	27.60
	26.29	9.45	26.58	23.66	27.89	27.26
	27.62	9.80	26.77	23.77	26.01	26.58
	27.20	9.90	25.38	24.70	26.69	26.42
		10.18		24.54	25.49	24.72
STD	0.97	0.26	0.70	0.46	1.01	1.11
Mean	27.44	9.82	26.41	13.42	22.27	22.10

STD/Mean %	3.54%	2.67%	2.65%	3.42%	4.53%	5.04%
------------	-------	-------	-------	-------	-------	-------

Table. Statistics on the diameters. The fluctuation in the diameters size is ranging between approximately 3 and 5 %.

2- Unfortunately, embedding the sample in a matrix cannot be done because it would destroy with capillary forces the suspended DNA bundles but, even more importantly, it would separated the helices losing the bundle configuration.

Finally, as we stated in the paper the main focus of our work is to demonstrate the biosensing device capability and its potentiality to evaluate variations in intercalants administration. Up to now the frequency splitting is only measurable with vibrometric technique in vacuum and, even if not fundamental for this work (Young's modulus for optimal intercalant dose determination), it represents a positive outcome that reveals nanostructural information difficult to directly measure with other techniques.

Reviewer 3

Q2: Somewhat addressed. The motivation from a pure knowledge perspective is now more clear, but from an application perspective is still somewhat murky. See Reviewer 2, Q2 and Reviewer 1, Q4

A2: According to reviewers suggestion, we expanded the present manuscript adding titration experiments of pristine DNA exposed to different concentration of the chemoterapeutic compound CisPt. We think that the titration experiments added to this version of the manuscripts strengthen our statements on the possible use of this technique for drug dosing. These newly added experiments demonstrate the feasibility of this application.

Q3: Please see the following references:

doi.org/10.1002/mame.201800302

doi.org/10.1039/B717581G

doi.org/10.1116/1.2801886

doi.org/10.1021/nl203503s (has some good references in the DNA origami area)

A3: We thank the reviewer for the suggested references. We added a comparison with other work on nanofibers presenting values of elastic modulus. We found that our findings are in line with the other works in literature, which range from 1 to 15 GPa. Please notice that a brief discussion on the DNA-Origami technique and related suggested references has been reported in Answer 1 to Reviewer 2 and in the text (see answer above).

Q6: The addition of R^2 values helps tremendously. Addressed, except perhaps for CisPT...the R^2 values here are not so different, why is that?

A6: The possible explanation is related to the way the CisPt deform the structure. While YOYO and GelRed intercalated into the helix, CisPt causes a stronger alteration of the double helix, which can insert a higher variability of the geometrical and structural properties. Such variability is also reflected in higher relative error of CisPt Young's modulus with respect to other samples. Probably for this reason, the two R^2 values are both relatively low and not so different.

Q8: While the proposed experiment is admittedly not trivial, it should be possible to image the bundle cross section. Low viscosity embedding resins exist, and even if the bundle were detached during embedding, would that mean that the eccentricity would be lost? The authors themselves, by frequency analysis, suggest that the tensile stress in the suspended bundles is quite small, so it isn't clear that detaching would be a major issue. The only issue that may admittedly be impossible to overcome perfectly would be ensuring the section is perfectly perpendicular to the bundle axis; this could render the sectioning approach meaningless. In any case, some secondary verification that the peak splitting is due to eccentricity would make this claim stronger.

A8: Comments to the resin embedding and cross section, TEM imaging related to diameters fluctuations and sample tilting, have been reported in Question 3 of the Review to Reviewer #2 raised by the Reviewer #3.

This reviewer agrees that eccentricity is certainly the most likely explanation, but a visual verification would be nice, particularly as it was given previously but withdrawn.

A8: Related to the withdrawn of visual verification, the authors would like to point out that in Lines 208-210 of the first manuscript version was written: “*Consequently, the splitting of the resonance peak in Figure 4d evidences a deformation of the circular cross-section towards an elliptical shape in our DNA resonators, **barely observable with electron microscopy analysis***”. This sentence was not referred to any SEM image (Figure 4d is a graph showing the splitting of the peaks) but it was referred to the difficulty in evaluating such a variation with a SEM and with a TEM as reported in the previous answers to the reviewers. The concept was actually improperly stated in that version, for this reason the sentence (actually no figure was shown to support the statement above) has been removed to avoid any confusion, since we have not observed this asymmetry with EM analysis. However, the mechanical analysis clearly shows the splitting of the resonance peaks, which is a phenomenon due to the presence of two inertia momentum (and so an elliptical section in our case), as reported in other literature works (10.1038/nano.2016.189, 10.1038/nano.2010.151, 10.1063/1.4813819, 10.1088/0957-4484/22/45/455502).

Q12: The authors are now saying that the bundles are not stable under either TEM or laser...but aren't they using a laser to measure them? And a TEM to image them? What are the issues with respect to stability under a laser, and does that affect the results in this manuscript?

A12: The comment in the previous answer was not related to the DNA bundle, but to single DNA double helix: “*Notice also that we produced suspended single DNA*”

filament but its mechanical stability, under TEM and laser beams, is poor and it is difficult to have reproducible measurements for our mechanical and structural study." To be clear, the stability is an issue only when a single double strand DNA is used and not for bundles. We added this comment because the molecular dynamic simulations were performed on single DNA filament and not on the bundles, even if the trend in the mechanical properties is the same. The DNA bundles are stable both under laser and TEM in the experimental conditions used in this work. Instead the single filaments are not stable and therefore they have not been reported in the manuscript, because it was not possible to make reproducible measurements.

Q13: This answer suggests that the authors believe that, in order for a bundle to be formed spanning two pillars, the length of the DNA molecules composing the bundle must be at least the pillar-pillar distance...do the authors have proof for this? Are they suggesting each molecule starts and stops on a pillar? What if only 50% of the molecules were broken?

A13: To clarify Answer 13 to Reviewer #3 of the first revision, we must add that if the presence of saturating (1x) and oversaturating (2x, 4x) amounts of chemotherapeutic compound induces double strand breaks (DSBs), this would lead to the formation of extremely short fragments that cannot be suspended and imaged with this pillar-pillar interdistance. In fact, past control experiments with solutions of short DNA fragments (range 1000-8000 bp) didn't show any evidence of bundles formation with the present inter-pillar distance.

According to HRTEM imaging and vibrometric measurements as a function of titration we don't see any behavior variation of inter-pillar bundle suspension with respect to pristine DNA so we believe that the DNA/intercalant molecules bridges between the pillars ("start and stops on a pillar").

The argument that only a fraction (like 50%) of the molecules can be affected by DSBs cannot be sustained since we have tested strongly oversaturated solutions which did not led to fragmentation as proved by the formation of whole bundles.

In the DISCUSSION and OUTLOOKS section we added the phrase *"In the work herein presented, the HRTEM direct imaging suggests that the main molecular variations occurring in the double helices after platinum drugs intercalation are structural modifications and not related to double strands breaks. Double strand breaks would produce much shorter DNA fragments that cannot be suspended and imaged with this pillar-pillar interdistance, as confirmed by control experiments (data not shown)."*

A minor comment:

1) The plots at the bottom left of the movies are impossible to read, the font for the axes labels and numbers is too small and so it isn't clear from watching the movie what the plot is meant to indicate.

A For better clarity we expanded the caption of the movies in the SI, giving all the information needed to read the plots correctly. The following statements were added *"In the graphs, the abscissa shows the time of simulation (from 0 to 20 ns), while the ordinate shows the DNA elongation (in the range 0 to 120 Å for the DNA and DNA-*

YOYO-1 plots, and 0 to 160 Å for the DNA/CisPt plot). The graphs develop over the movie time, in sync with the portrayed DNA elongation."

Several of the initial issues with this manuscript were addressed, yet there remain some lingering issues that are pretty substantial. The one which all 3 reviewers seemed to have concerns about is how this technique would be used in the applications described; I do not think the authors have clearly stated this, and the simulated titrations do not do much to convince the reader of the utility. While the argument that bundle eccentricity causes peak splitting is reasonable and likely true, it would be more convincing if some additional evidence were provided for this....particularly since the image supposedly showing this evidence was removed because it didn't really do so.

A We would like to thank the Referees for the accurate reading and suggestions. We believe that we have now addressed extensively all the questions raised and provided additional experiments to clarify and expand the findings and motivation of the present work. In particular, we accurately designed and performed the titration experiments that now show a clear titration curve for CisPt. We explored different options to directly measure the eccentricity, but we verified that HRTEM approach is limited for this task. We are not aware of any removal from the previous versions of the manuscript of images claiming evidence of eccentricity.

REVIEWERS' COMMENTS:

Reviewer #2 (Remarks to the Author):

This version of revised manuscript has provided new evidences to support the value of this DNA bundle resonator system in the application of DNA intercalation molecule sensing. The responses from the authors acknowledged the facts and addressed reviewers concerns sincerely and properly. I recommend accepting of this paper for publication in Nature Communications.

Reviewer #3 (Remarks to the Author):

The addition of the titration data improves the impact of this work substantially. The data in figure 5d suggest there may be issues relating to accuracy of the proposed dosage measurements, due to the large error bars on the calibration curve (i.e. how well could this technique really determine dosage quantitatively, with a such a calibration curve?). However, this concern could certainly be addressed in future work, and I don't believe this should prevent publication at this stage in the technology (though it might be nice to have a sentence addressing this in the current manuscript).

The quality of the writing could be improved a little bit, as there are minor grammar issues scattered throughout the manuscript, but overall it is clear.

Overall, I believe this work is currently suitable for publication.

Authors' response to the reviewers' comments:

Dear Editor,

We would like to thank you for the acceptance of the manuscript and all reviewers for positive remarks made after the last revision.

Please find below the *point by point* answers to the last reviewer comments.

Reviewers' comments:

Reviewer #2 (Remarks to the Author):

This version of revised manuscript has provided new evidences to support the value of this DNA bundle resonator system in the application of DNA intercalation molecule sensing. The responses from the authors acknowledged the facts and addressed reviewers concerns sincerely and properly. I recommend accepting of this paper for publication in Nature Communications.

A. We thank the reviewer for the positive comment.

Reviewer #3 (Remarks to the Author):

Q1. The addition of the titration data improves the impact of this work substantially. The data in figure 5d suggest there may be issues relating to accuracy of the proposed dosage measurements, due to the large error bars on the calibration curve (i.e. how well could this technique really determine dosage quantitatively, with a such a calibration curve?). However, this concern could certainly be addressed in future work, and I don't believe this should prevent publication at this stage in the technology (though it might be nice to have a sentence addressing this in the current manuscript).

A1. We thank the reviewer for the positive comment. We are aware that the error bars in the calibration curve are large, but the titration experiment in this manuscript was conducted as a proof of concept of the possibility to distinguish different intercalant concentration. In order to exploit this technology, further improvements on the bundles uniformity and stability should be performed, but it would be the aim for future works. We added a sentence in the Discussions section regarding this comment:

“We notice that the experimental error in figure 6d can further be reduced. In future work we will make an effort to improve the suspension process control conditions, such as temperature, humidity and buffer composition, in order to obtain a higher number of bundles with higher uniformity in size and composition.”

Q1. The quality of the writing could be improved a little bit, as there are minor grammar issues scattered throughout the manuscript, but overall it is clear.

Overall, I believe this work is currently suitable for publication.

A1. We revised the manuscript in order to correct grammar error and improve the writing